# Retrieval of stratospheric aerosol extinction coefficients from sun-normalized OMPS-LP measurements

Alexei Rozanov[1], Christine Pohl[1], Carlo Arosio[1], Adam Bourassa[2], Klaus Bramstedt[1], Elizaveta Malinina[1,3], Landon Rieger[2], and John P. Burrows[1]

[1]Institute of Environmental Physics, University of Bremen, Germany
[2]University of Saskatchewan, Canada
[3]now at Canadian Centre for Climate Modeling and Analysis (CCCma), Environment and Climate Change Canada (ECCC), Canada

**Correspondence:** Alexei Rozanov (alex@iup.physik.uni-bremen.de)

**Abstract.** A new retrieval approach to obtain vertical profiles of the aerosol extinction coefficient from the measurements of the scattered solar light in the limb viewing geometry made by the OMPS-LP instrument is presented. In contrast to many other published limb-scatter retrievals, our new algorithm does not employ the normalization by a limb measurement at an upper tangent height. Instead, the measured limb radiances are normalized to the solar irradiance. The main advantage of this approach is an almost complete elimination of the dependence of the retrieval results on the prior aerosol extinction profile used in the retrieval. This makes the retrieval better suited for the analysis of the observation scenes with highly elevated aerosol plumes as occurred after the Hunga Tonga–Hunga Ha'apai volcanic eruption in January 2022. The results from the new approach were compared to the vertical profiles of the aerosol extinction coefficients retrieved from SAGE III/ISS and OSIRIS. In general, an agreement within 25% between the different data products was observed in the 18 - 23 km altitude range although larger differences were seen after very strong volcanic eruptions and wildfires. In comparison with OSIRIS, larger differences are seen at southern high latitudes (above 60°S). The new data product was used to investigate the evolution of the aerosol plume after the Hunga Tonga–Hunga Ha'apai volcanic eruption.

## 1 Introduction

Stratospheric aerosols directly influence the radiative budget of the Earth's atmosphere by scattering the incident solar radiation back to the space in the UV-Vis-NIR spectral range and by absorbing the radiation upwelling from the troposphere in the thermal infrared spectral range. As discussed, e.g. by Solomon et al. (2011), the presence of an increased amount of aerosols in the stratosphere leads to a global cooling. The cooler temperatures increase the rate of formation of ozone ($O_3$) (Groves et al., 1978; Groves and Tuck, 1979) and probably decrease the rate of catalytic removal of $O_3$. In addition, stratospheric aerosols serve as nuclei for the formation of the polar stratospheric clouds (PSCs) or directly provide surface for heterogeneous

reactions releasing photo-labile halogens ($Cl_2$, $Br_2$ and $I_2$, but noting that only small amounts of I reach the stratosphere) and inter-halogens ( BrCl, ICl and IBr), which are photolyzed and yield halogen atoms. These react with $O_3$ generating halogen oxides. Aerosols also take up $HNO_3$ reducing the amount of $NO_x$. At sufficiently low $NO_x$, the ClO disproportionation reaction and the reaction of BrO with ClO participate in additional catalytic cycles, which leads to the formation of the ozone holes in the polar regions (Solomon et al., 1986; Portmann et al., 1996; Tritscher et al., 2021). At high aerosol loading, similar processes might become significant also at extra-polar latitudes. For example, a remarkable ozone loss of up to 18 Dobson units was identified in the southern mid-latitudes after the 2020 Australian wildfires (Solomon et al., 2023) while (Evan et al., 2023) reported a decrease in the stratospheric $O_3$ above the tropical southwestern Pacific and Indian Ocean region by 5% after the 2022 Hunga Tonga–Hunga Ha'apai eruption. However, as discussed by (Zhu et al., 2023), the latter decrease was mainly caused by the injection of a large amount of water vapour with injection of ClO, which led to the chlorine activation in the heterogenic reactions. Thus, the ozone decrease reported by (Evan et al., 2023) seems to be specific to the Hunga Tonga–Hunga Ha'apai eruption. Besides the temperature and stratospheric ozone amount, the global precipitation, sea level pressure and circulation patterns, El Niño and climate extremes were reported to be affected by volcanic eruptions, see e.g. (Fyfe et al., 2013; Khodri et al., 2017; Paik and Min, 2018).

Despite its high scientific importance, the availability of information about the stratospheric aerosols on the global scale is quite limited. One of the widely used characteristics of the stratospheric aerosol, which is available from several space-borne instruments currently in operation, is the stratospheric aerosol extinction coefficient. As discussed by several authors, this quantity can be used to calculate the radiative forcing due to stratospheric aerosols, see e.g. (Hansen et al., 2005; von Savigny et al., 2015; Kloss et al., 2020; Malinina et al., 2021). Global vertical distributions of the stratospheric aerosol extinction coefficient can only be retrieved from space borne measurements. Most robust are the data from solar occultation measurements as measured spectra are self-calibrated through exo-atmospheric measurements and the corresponding retrievals do not use any strong assumptions in the forward modeling. Such measurements, for example, were performed quasi-globally since 1979 by the SAGE instrument series (McCormick, 1987; McCormick et al., 1989; Chu et al., 1997). Data products from occultation instruments are well suited for the use as a validation source but suffer from a relatively sparse spatial sampling and coverage. Much more dense spatial sampling is provided by the aerosol extinction data retrieved from measurements of the scattered solar light in limb viewing geometry made by OSIRIS (Llewellyn et al., 2004), SCIAMACHY (Burrows et al., 1995; Bovensmann et al., 1999) and OMPS-LP (Flynn et al., 2014) instruments. In contrast to the solar occultation measurements, the retrieval of the aerosol extinction coefficients from limb-scatter measurements requires assumptions about the aerosol particle size distribution and aerosol amount around the normalization tangent height, if this kind of the normalization is applied, see e.g. (Rieger et al., 2015). Furthermore, the limb-scatter radiance strongly depends on the reflectance of the troposphere and surface. In the nadir viewing geometry, the aerosol extinction coefficient is retrieved from active remote sensing measurements performed by lidar instruments, such as CALIOP (Winker et al., 2009). Lidar retrievals also employ normalization to measurements at high altitudes resulting in a sensitivity to the aerosol at normalization altitude. The major source of uncertainty for lidar instrument is a lack of the knowledge of the lidar ratio, see e.g. (Young et al., 2013).

This study focuses on the retrieval of the aerosol extinction coefficient from space borne limb-scatter measurements. Although several algorithms for such retrievals have been successfully developed (Bourassa et al., 2012; von Savigny et al., 2015; Rieger et al., 2019; Malinina et al., 2021; Taha et al., 2021; Bourassa et al., 2023) all of them are optimized to mitigate the influence of the surface reflectance and some also minimize error arising from the usage of a fixed aerosol particle size distribution. In this study we present a new algorithm to retrieve vertical distributions of the aerosol extinction coefficient from

limb-scatter measurements made by the OMPS-LP instrument. The algorithm has been developed at the University of Bremen. One of its objectives is to minimize the influence of the unknown aerosol loading at high altitudes. This is especially crucial in the case of volcanic eruptions that reach high altitudes, such as the Hunga Tonga–Hunga Ha'apai in January 2022. The latter was the strongest volcanic eruption in the $21^{st}$ century with a highest volcanic plume observed in the satellite era and injected $0.4 - 0.9$ Tg of $SO_2$ to altitudes up to $\sim$32 km and a huge amount of the water vapor to altitudes up to $\sim$50 km, see e.g. (Carn

et al., 2022; Carr et al., 2022; Duchamp et al., 2023; Legras et al., 2022; Millán et al., 2022; Proud et al., 2022; Xia et al., 2024).

      In addition to the OMPS-LP retrieval developed at the University of Bremen and described in this study, algorithms from NASA (Taha et al., 2021) and the University of Saskatchewan (Bourassa et al., 2023) also exist. An intercomparison of the extinction profiles generated by the three algorithms is not straightforward due to the different assumptions about particle size

distribution used in the NASA retrieval and the tomographic approach employed in the retrieval developed by the University of Saskatchewan. This intercomparison is a subject of a dedicated study, which is currently ongoing. The CALIOP aerosol extinction profile data (Young et al., 2018) is also available but not used in this study. This is because a very different measurement principle, much smaller footprint in the cross track direction resulting in a significantly sparser coverage, uncertainty from the lidar ratio, and typically low signal to noise ratio in the stratosphere would make the interpretation of the obtained results and

attribution of the differences much more ambiguous as compared to the selected datasets (OSIRIS and SAGE III). CALIOP data might be extremely valuable for comparisons exercises dedicated to the latitudes, where SAGE III data are not available (e.g. winter high latitudes) or regions where a discrimination between the aerosol types or between aerosol and cirrus clouds is essential (e.g. tropical UTLS). Detailed comparisons of this kind are, however, outside the scope of this study.

      The manuscript is structured as follows. In Sect. 2 the observational data used for the retrieval and comparisons are described.

Section 3 presents a detailed description of the retrieval algorithm. The selection of the retrieval wavelength is justified in Sect. 4. The main advantages of the new retrieval algorithm are discussed in Sect. 5. Comparisons with independent data are discussed in Sect. 6. Application of the new data set for tracing the evolution of the aerosol plume after the Hunga Tonga–Hunga Ha'apai eruption is presented in Sect. 7. Conclusions are presented in Sect. 8.

## 2   Measurement data

The Ozone Mapping and Profiler Suite (OMPS) was launched in October 2011 onboard the Suomi NPP satellite, a joint venture of NOAA/NASA (Flynn et al., 2014). The satellite flies in a sun-synchronous orbit with 98.78° inclination at the altitude of about 800 km. OMPS comprises three instruments: the Nadir Mapper (NM), the Nadir Profiler (NP) and the Limb Profiler (LP).

In this study, only the measurements from the latter instrument, OMPS-LP, are used. The OMPS-LP instrument observes solar radiance scattered by the Earth's atmosphere including that passing through the atmosphere downwards and reflected back by the underlying surface. The radiance from the atmosphere enters the instrument through three slits pointing in the direction opposite to that of the satellite's flight. By using two-dimensional charged-coupled device (CCD) detectors, measurements are made simultaneously in the altitude range from approximately 0 to 100 km with a vertical sampling of 1 km. Because of unresolved calibration issues, only the measurements from the central slit are used in this study. OMPS-LP instrument is a prism spectrometer covering the 280 - 1000 nm spectral range with a spectral resolution increasing with the wavelength from 1 nm in the UV to about 30 nm in the near-IR region. The vertical field of view of each detector pixel is about 1.5 km. In this study, the Level 1G data V2.6 provided by NASA are used. The dataset contains both sun-normalized and unnormalized radiances.

The Stratospheric Aerosol and Gas Experiment (SAGE) III instrument has been operating since February 2017 onboard the International Space Station (ISS). The ISS flies at ∼400 km altitude and its orbit has an inclination of 51.6°. SAGE III/ISS instrument performs solar occultation measurements during sunrise and sunset. The aerosol extinction coefficients are provided at 9 wavelengths: 384, 449, 520, 602, 676, 756, 869, 1021, and 1544 nm. The retrievals cover the range from the cloud top to about 45 km altitude at a vertical resolution of about 0.75 km. In this study Level 2 V5.3 data from the SAGE III/ISS instrument were used. No cloud filtering was applied. Negative values in SAGE III/ISS data were not filtered out to avoid creating a positive bias.

The Optical Spectrograph and InfraRed Imaging System (OSIRIS) instrument (Llewellyn et al., 2004) was launched in November 2001 onboard the Swedish Odin satellite. The satellite flies in a sun-synchronous orbit with 98.78° inclination at the altitude of about 600 km. The instrument combines a grating UV/Vis spectrograph and an IR imager and observes the solar radiance scattered by the Earth's atmosphere including that reflected by the underlying surface. The observations are done in the altitude range between about 0 and 100 km with a vertical sampling of ∼2 km and similar vertical resolution. The optical spectrograph, whose data were used in this study, measures the spectral radiance in the 280–800 nm range with a resolution of 1–2 nm. OSIRIS retrieval provides vertical profiles of the stratospheric aerosol extinction coefficient at a wavelength of 750 nm between about 8 and 40 km with a vertical resolution of about 2 km. In this study, OSIRIS cloud cleared Level 2 data of V7.3 were used (Rieger et al., 2019). Because of issues identified by the University of Saskatchewan team in the vertical profiles of the aerosol extinction coefficient retrieved after the Hunga Tonga–Hunga Ha'apai eruption, only data before January 2022 are considered.

## 3 Retrieval algorithm

In this section, a detailed description of the new OMPS-LP retrieval algorithm of the University of Bremen (V2.1) is presented. The retrieval approach is based on a regularized non-linear inversion[1]. The algorithm is similar to those employed for the retrieval of vertical distributions of atmospheric species by the University of Bremen team and many other authors, e.g. Glatthor

---

[1]One of the popular techniques of this type is the maximum a posteriori information approach described by Rodgers (2000)

et al. (2006); Livesey et al. (2006); Rozanov et al. (2011); Malinina et al. (2018); Rieger et al. (2019); Mettig et al. (2021); Keppens et al. (2024). This kind of the algorithms is also referred to as the global fit approach because all available spectral data are fitted together at once and no separation into spectral and vertical steps is done.

In general, the following quadratic form needs to be minimized to obtain the solution (see, e.g. Rodgers (2000)):

$$\| y - F(x_o) - K(x - x_0) \|_{S_y^{-1}} + \| x - x_0 \|_{S_r}, \tag{1}$$

where $x$ is the resulting state vector, i.e. the solution to be obtained, $x_0$ is the initial guess (a priori) state vector, $K = \frac{\delta F(x)}{\delta x}\Big|_{x=x_0}$ is the Jacobian of the forward model operator also referred to as the weighting function, $y$ is the measurement vector constructed from the observed values, $F(x_0)$ is the measurement vector simulated by the forward model assuming the initial guess atmospheric state, $S_y^{-1}$ is the noise covariance matrix and $S_r$ is the regularization matrix.

The state vector contains aerosol extinction coefficient values at altitude levels corresponding to the measurement tangent
heights and the effective Lambertian surface albedo. The initial guess values for the aerosol profile are set in accordance with an arbitrary profile extracted from the SAGE II aerosol climatology published by Bingen et al. (2004). As the retrieval depends only weakly on the initial guess profile (see below), selection of any particular profile is rather unimportant. For the effective surface albedo, the initial guess value is set to 0.5. The initial guess state vector is the same for all retrievals. The aerosol extinction coefficient and the scattering phase function are calculated employing Mie theory assuming a spherical shape of
aerosol droplets. Optical properties of the aerosol particles are taken from the OPAC data base (Hess et al., 1998) for the "sulfate droplets" aerosol type. This aerosol type corresponds to background stratospheric aerosols (75% solution of $H_2SO_4$). The calculations are made for the relative humidity of the ambient air of 0%. A unimodal log-normal particle size distribution (PSD) with fixed parameters (median radius, $R_{med}$, of 0.08 $\mu$m and geometrical standard deviation, $\sigma$, of 1.6) is assumed for all calculations. The PSD parameters are taken from an arbitrary balloon-borne in-situ measurement for background aerosol
conditions reported by Deshler (2008). An aerosol layer between 0 and 50 km and an aerosol free atmosphere above are assumed.

The measurement vector contains logarithms of the limb radiance in the tangent height range 8.5 – 48.5 km (with 1 km sampling) normalized by the solar irradiance. This normalization approach is the main difference between the new retrieval and the previous version of the OMPS-LP retrieval of the University of Bremen (V1.0.9) published by Malinina et al. (2021),
where the normalization used a limb measurement at an upper tangent height. As the normalization by the solar irradiance instead of the reference tangent height makes the retrievals more sensitive to the surface reflectance, the approach to retrieve the effective Lambertian surface albedo was optimized as follows. In the new retrieval, the effective surface albedo is included in the state vector and retrieved simultaneously with the aerosol extinction coefficients. Measurements at all tangent heights in 8.5 – 48.5 km range are used to retrieve the effective surface albedo. In contrast, the V1.0.9 approach performed alternating
independent retrievals of the aerosol extinction coefficients and of the effective surface albedo. The latter retrieval used only the limb measurement at the reference tangent height. In addition, V2.1 approach is more sensitive to the quality of the radiative transfer modeling. For this reason, the fully spherical mode of the SCIATRAN radiative transfer model (Combined Differential-Integral approach involving the Picard Iterative approximation, CDIPI) was used for the forward modeling rather than the

**Table 1.** Retrieval settings in OMPS-LP V2.1 and V1.0.9 algorithms of the University of Bremen.

|  | Old retrieval version: V1.0.9 | New retrieval version: V2.1 |
| --- | --- | --- |
| Retrieval wavelength | 869 nm | 869 nm |
| Tangent height range | 12.5 – 37.5 km | 8.5 – 48.5 km |
| Normalization approach | Limb measurement at the normalization tangent height (37.5 km) | Solar irradiance |
| Retrieval approach | Regularized non-linear inversion | Regularized non-linear inversion using Levenberg-Marquardt approach |
| Regularization | Zeroth and first order Tikhonov regularization w.r.t. the solution at a previous iterative step | Zeroth and first order Tikhonov regularization w.r.t. the solution at a previous iterative step |
| Radiative transfer model | SCIATRAN, approximate spherical mode | SCIATRAN, full spherical mode |
| Assumed aerosol layer | 12 – 45 km | 0 – 50 km |
| Assumed aerosol PSD | unimodal log-normal distribution, fixed parameters: $R_{med} = 0.08\ \mu$m, $\sigma = 1.6$ | unimodal log-normal distribution, fixed parameters: $R_{med} = 0.08\ \mu$m, $\sigma = 1.6$ |
| Aerosol/albedo retrieval | Independent at each step, alternated | Joint, simultaneous |
| Measurements used for the surface albedo retrieval | Limb measurement at the normalization tangent height (37.5 km) | All tangent heights included in the retrieval (8.5 – 48.5 km) |

approximate spherical mode (Combined Differential-Integral approach, CDI) used in V1.0.9 algorithm. A detailed description of the CDI and CDIPI modes of SCIATRAN can be found in (Rozanov et al., 2000, 2001). We note that the use of the solar normalization makes the retrieval more sensitive to the quality of the absolute calibration of the measurements. In particular, an error can be expected if the absolute calibration of the limb spectra and that of the solar irradiance spectra differ. Although absolute calibration errors are typically independent of the incoming signal, they might be differently re-distributed by the retrieval between the retrieved parameters (aerosol extinction coefficient and effective surface albedo in our case) depending on the observational geometry and surface reflectivity. This makes them difficult to be identified. In a case of a degradation of the detector a drift in the data might be expected. A degradation of detectors is, however, more an issue in the UV spectral range rather than in near-IR used in this study. For now, no indications for any issues in the absolute calibration of OMPS-LP instrument or degradation of the detectors are identified.

The retrieval is done for the relative changes in the aerosol extinction coefficient, $([x]_i - [x_0]_i)/[x_0]_i$, and absolute changes in the effective surface albedo, $[x]_j - [x_0]_j$. In the formulas above, the square brackets denote the componentwise operation, $i$ runs through all aerosol levels included into the retrieval and $j$ is the number of the effective surface albedo component in the state vector. The regularization is based on the Tikhonov approach with zeroth and first order terms:

$$S_r = S_a^{-1} + \gamma^{-2} S_d^T S_d. \tag{2}$$

The zeroth order Tikhonov term, $S_a^{-1}$, acts in a similar way as the a priori covariance matrix in the maximum a posteriori method described by Rodgers (2000). The diagonal elements of the matrix $S_a$ are set to 0.3 for the aerosol extinction coefficients and to 0.01 for the effective surface albedo. The off-diagonal elements are selected assuming a 1 km correlation radius

for the aerosol extinction coefficients and no correlation between the aerosol extinction and the effective surface albedo. The matrix $S_d$ in the first order Tikhonov term is the first derivative matrix. Detailed formulas to calculate the elements of the regularization matrices can be found in (Rozanov et al., 2011). The Tikhonov parameter, $\gamma$, is set to 0.2. The noise covariance matrix is a diagonal matrix with diagonal elements corresponding to a signal-to-noise ratio of 200, independent of the tangent height. This value as well as the values of Tikhonov parameters were selected empirically as a trade-off between the stability and sensitivity of the retrievals. Taking into account that the regularization strength is determined by the ratio of the noise covariance and Tikhonov parameters rather then their absolute values, we prefer to use an empirical value for the signal-to-noise ratio rather than the reported measurement errors. This keeps the regularization strength similar for all retrievals.

The non-linearity of the inverse problem defined by Eq. 1 is accounted for by employing an iterative scheme. The Levenberg-Marquardt algorithm is used to obtain the final solution. At each iterative step, the solution of the non-linear inverse problem is written as (see, e.g. (Rodgers, 2000) for details):

$$x_{i+1} = x_i + \left[K_i^T S_y^{-1} K_i + S_r + \lambda S_a^{-1}\right]^{-1} K_i^T S_y^{-1} \left(y - F(x_i)\right), \tag{3}$$

where $i$ is the iteration number and $\lambda$ in the Levenberg-Marquardt parameter with an initial guess of 1. We note that the regularization is done with respect to the solution at the previous iterative step rather than with respect to the initial guess atmospheric state. With that, a constraint to the magnitude and shape of the difference profile is applied only within a single iterative step making the difference between the final solution and the initial guess state much less constrained. The iterative process stops if the aerosol extinction coefficient at two subsequent iterative steps does not change by more than 2% between 15 and 28 km or if the relative change in the total root mean square difference is below 0.001. The convergence criteria are selected empirically taking into account the targeted precision of the retrieved aerosol extinction profiles of about 10%. The results are rejected if the retrieval requires more than 100 iterations to converge. About 80% of the retrieval runs converged within 17 iterations (histograms of the number of iterations needed for the convergence are presented in Fig. S1 in the supplement).

The averaging kernels, which describe the sensitivity of the retrieval to the true state, are calculated at the last (n-th) iteration by using the following formula:

$$A = \left[K_{n-1}^T S_y^{-1} K_{n-1} + S_r + \lambda S_a^{-1}\right]^{-1} K_{n-1}^T S_y^{-1} K_{n-1}. \tag{4}$$

The most important settings used in the new OMPS-LP retrieval approach of the University of Bremen (V2.1) and those of its precursor version (V1.0.9) are summarized in Table 1.

## 4 Optimization of the retrieval wavelength

The optimal wavelength for the OMPS-LP V2.1 aerosol retrieval of the University of Bremen is selected by (i) analyzing possible contamination of the measured signal by interfering spectral signatures of other atmospheric constituents and (ii) investigating the sensitivity of limb-scatter measurements to the vertical distribution of the stratospheric aerosol.

To investigate the potential influence of interfering absorbers, a measured or modeled atmospheric spectrum, which has higher spectral resolution and sampling than that of OMPS-LP, is required. To ensure that all relevant spectral signatures from

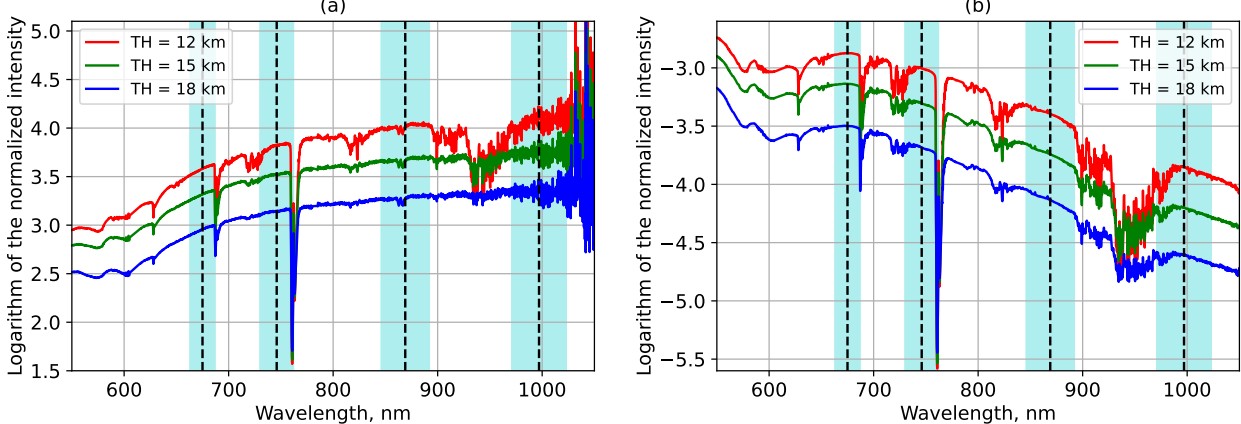

**Figure 1.** Spectra of limb-scatter radiance at different tangent heights normalized by the limb measurement at the reference tangent height (panel (a)) and by the solar irradiance spectrum (panel (b)). Dashed lines depict selected central wavelengths used by the NASA aerosol retrieval (675, 745, 869 and 997 nm). The cyan shadings mark ranges of $\pm$ FWHM of the OMPS-LP instrument around the central wavelengths of the NASA aerosol retrieval.

all atmospheric constituents are taken into account and their strength is representative for real observational conditions, we decided to analyze a measured spectrum rather than the modeled one. A suitable spectral information was provided by the SCIAMACHY spectrometer, which flew onboard the European Envisat satellite from March 2002 to April 2012 and had a similar observational geometry to that of the OMPS-LP instrument. While the most recent version of the University Bremen retrieval described in this study uses the sun-normalized limb-scatter measurements, all previous retrieval versions as well as most of the other limb-scatter stratospheric aerosol retrievals use the limb radiances normalized by the limb measurements at an upper tangent height (also referred to as the reference tangent height). For this reason, below we analyze the spectral behavior of both the sun-normalized and reference tangent height normalized radiance.

Figure 1 shows spectra of the limb-scatter radiance at different tangent heights normalized by the limb measurement at the reference tangent height of 35 km (panel (a)) and by the solar irradiance spectrum (panel (b)). The spectra were obtained from the SCIAMACHY measurement performed on January $7^{th}$, 2004 at 16:24:37 UTC with the tangent point ground coordinates 26°S, 5°W. The vertical dashed lines mark the wavelengths of 675, 745, 869 and 997 nm, which were used by the NASA OMPS-LP V2.0 stratospheric aerosol extinction retrieval (Taha et al., 2021). The cyan shadings mark the areas smeared by the spectral response function (SRF) of the OMPS-LP instruments. The width of the shadings is selected as $\pm$ FWHM (full width at half maximum of the SRF). The latter were estimated from Fig. 1-3 in (Rault et al., 2010) as 12, 16, 23 and 26 nm at 675, 745, 869 and 997 nm, respectively. The figure reveals that the band centered at 675 nm touches at its long-wave edge the $O_2$-B absorption band centered at 688 nm. A similar situation is observed for 745 nm. At this wavelength, the measurements might be affected by the strong absorption in $O_2$-A band centered at 762 nm. A potential influence of the ozone absorption is similar for radiances normalized by the measurement at the reference tangent height and by the solar irradiance spectrum. The band

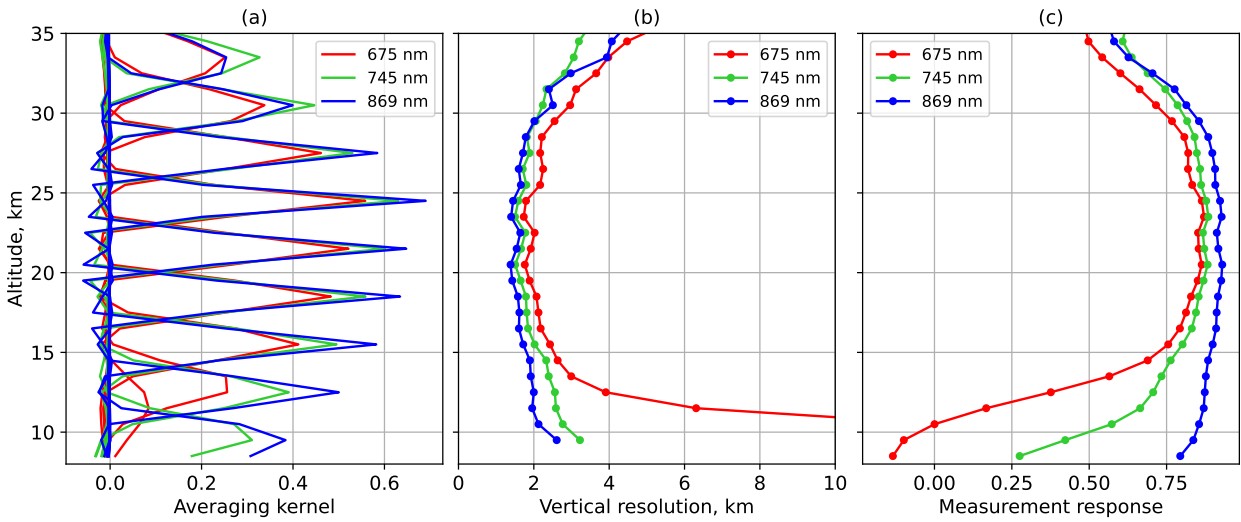

**Figure 2.** Characteristics of the OMPS-LP V2.1 aerosol extinction coefficient retrieval of the University of Bremen for different wavelengths. Panel (a): averaging kernels. Panel (b): vertical resolution. Panel (c): measurement response.

centered at 869 nm approaches at its long-wave edge the water vapor absorption band, however the potential interference is rather small. It might be advantageous to slightly shift the wavelength band towards the shorter wavelengths. However, due to

225 a sparse spectral sampling of the OMPS-LP Level 1 data, this is not possible without including the water vapor band on the short-wavelength side. For the reference tangent height normalization, a small spectral feature is present in the middle of the band, which is not present for the sun-normalized radiance. This is most probably an emission signature propagating from the measurement at the reference tangent height. At 997 nm, the measurements might be affected by the water vapor absorption at the short-wave edge of the band. This influence seems to be stronger for the sun-normalized radiance. The reason for that

is a partial canceling of the tropospheric signal when using the reference tangent height normalization. A strong increase of the measurement noise towards the longer wavelengths in the case of the reference tangent height normalization is typical for SCIAMACHY. For the OMPS-LP instrument, the signal-to-noise ratio at 997 nm is about a factor of two lower than that at 869 nm. Because of a stronger overlap with absorption features of atmospheric gases and lower signal-to-noise ratio as compared to other wavelengths, we exclude the 997 nm wavelength from the further consideration.

To investigate the sensitivity of the OMPS-LP V2.1 aerosol retrieval to changes in the vertical distribution of the aerosol extinction coefficient, the averaging kernels (see Eq. 4) at different wavelengths were analyzed for an example OMPS-LP measurement performed on January $6^{th}$, 2018 at 15:04:37 UTC with the tangent point ground coordinates 51°S, 14°W. The results are presented in Fig. 2. The averaging kernels (AKs) for wavelengths of 675, 745 and 869 nm are presented in panel (a) of the plot. For clarity reason, only every third AK between 9.5 and 33.5 km is plotted. It is seen that the AKs for 675

240 nm (red) are always smaller than those for 745 nm (green) and 869 nm (blue) and significantly degrade below 15 km. This is evidence of a lower sensitivity of the retrieval at 675 nm with respect to the other two wavelengths at all altitudes. At the

lowermost altitude shown in the plot (9.5 km), the AK for 675 nm is very small and its peak is strongly displaced from the nominal altitude, which is an indication of a nearly complete loss of sensitivity below 10 km. Comparing AKs for 745 and 869 nm, we see that the former are only slightly lower than those for 869 nm at almost all altitudes and even slightly higher above 30 km. For both wavelengths, AKs peak at their nominal altitudes.

Panel (b) of Fig. 2 shows the vertical resolution of the retrieval for different wavelengths. It is calculated as a reciprocal of the diagonal elements of the AK matrix multiplied by the step of the retrieval vertical grid (1 km in our case). Confirming the conclusions drawn from the left panel of the plot, the vertical resolution of the retrieval at 675 nm is only a bit lower compared to that at the two other wavelengths down to 13.5 km and starts to degrade rapidly below this altitude exceeding the value of 10 km at 11.5 km altitude level. The vertical resolution of the retrieval at 745 nm is slightly worse than that at 869 nm below 30 km and is slightly better above. A similar behavior is seen for the measurement response shown in panel (c) of Fig. 2 and calculated as a sum of the elements in the AK matrix rows. The measurement response for the retrieval at 869 nm is close to the ideal value of one with values better than 0.75 below 31.5 km altitude. At both 675 and 745 nm, the measurement response starts to degrade at lower altitudes crossing the 0.5 level at about 13 km for the former and about 10 km for the latter wavelengths.

From the discussion in this section we consider the wavelength of 869 nm to be optimal for the retrieval of vertical profiles of the stratospheric aerosol extinction coefficient from limb-scatter measurements performed by the OMPS-LP instrument.

## 5   Main advantages of the new retrieval algorithm

Several advantages are associated with the normalization of the limb radiance by the solar irradiance as it is done in the new V2.1 retrieval of the University of Bremen with respect to the normalization by the limb measurement at the reference tangent height as it was done in the precursor V1.0.9 retrieval described by Malinina et al. (2021) and other limb-scatter stratospheric aerosol retrievals (Bourassa et al., 2008; von Savigny et al., 2015; Taha et al., 2021). First, limb measurements at upper tangent heights generally have lower signal to noise ratios and are more contaminated by the stray light as compared to measurements at lower tangent heights. Thus, dividing all measurements by one at an upper tangent height generally degrades the quality of all data used in the retrieval. Second, a presence of an unknown signal from the aerosol scattering at the reference tangent height might bias the retrieval if the a priori aerosol content used in the forward model significantly differs from the real one. In addition, the reference tangent height is often used to estimate the effective Lambertian albedo of the underlying scene and a wrong assumption about the aerosol scattering contribution at this tangent height might bias the retrieved value of the effective albedo, which might additionally bias the retrieved vertical profile of the aerosol extinction coefficient.

The increase of the measurement noise when using the reference tangent height normalization approach is clearly seen on the example of SCIAMACHY measurements shown in Fig. 1. The spectra in panel (a) of the plot are obviously noisier compared to those in panel (b), especially for the longer wavelengths. Although, results from SCIAMACHY cannot be directly transferred to other instruments, a degradation of the measurement quality with an increasing tangent height is rather common for limb-scatter observations.

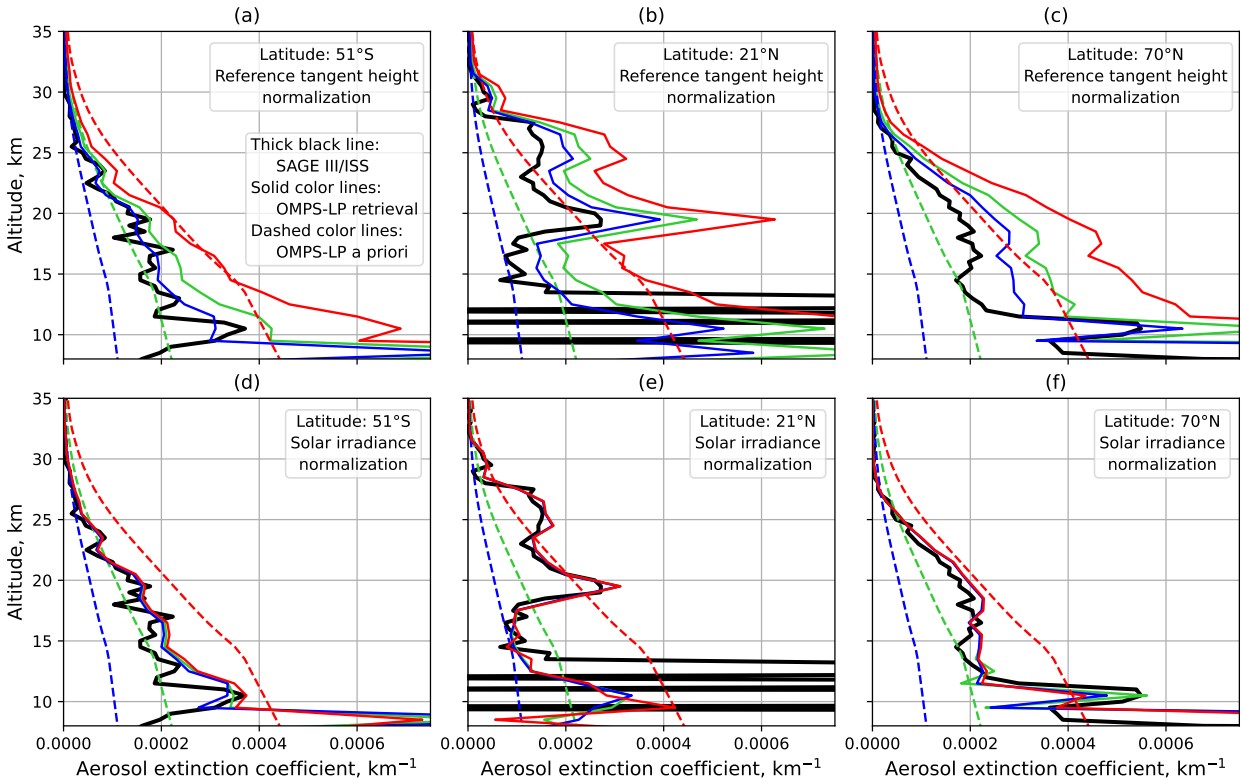

**Figure 3.** Dependence of the retrieval results on the a priori aerosol extinction profile for the University of Bremen retrieval using the normalization to the limb measurement at the reference tangent height (panels (a) – (c)) and to the solar irradiance spectrum (panels (d) – (f)). The comparisons are done for three example OMPS-LP measurements. Panels (a) and (d): January 6, 2018, 15:04:37 UTC; 51°S, 14°W. Panels (b) and (e): January 22, 2018, 10:20:45 UTC; 22°N, 43°E. Panels (c) and (f): July 25, 2018, 04:39:29 UTC; 70°N, 107°E.

To illustrate the dependence of the retrievals using different normalization approaches on the a priori aerosol extinction pro-
file, we performed retrievals for three example measurements of OMPS-LP using three different a priori profiles: the standard one, halved for all altitudes and doubled for all altitudes. The results of this investigation are shown in Fig. 3. We note that all results shown in this plot were obtained with V2.1 retrieval using either the normalization to the reference tangent height or to the solar irradiance, all other settings remained unchanged. The plot reveals that the results obtained using the normalization to the reference tangent height (panels (a) – (c)) strongly depend on the a priori profile used in the retrieval. We note that the bulk effect comes from the scaling of the a priori profile at the altitude level corresponding to the reference tangent height (results obtained by scaling the a priori profile only at and above the reference tangent height and only below it are shown in the supplement, Fig. S2). In contrast, the retrieval results obtained using the normalization to the solar irradiance (panels (d) – (f)) are almost independent from the a priori profile. In this case, small deviations between the results obtained for different a priori profiles are only seen below about 12.5 km altitude. The results for altitudes above 30 km are shown in Fig. S3 in the

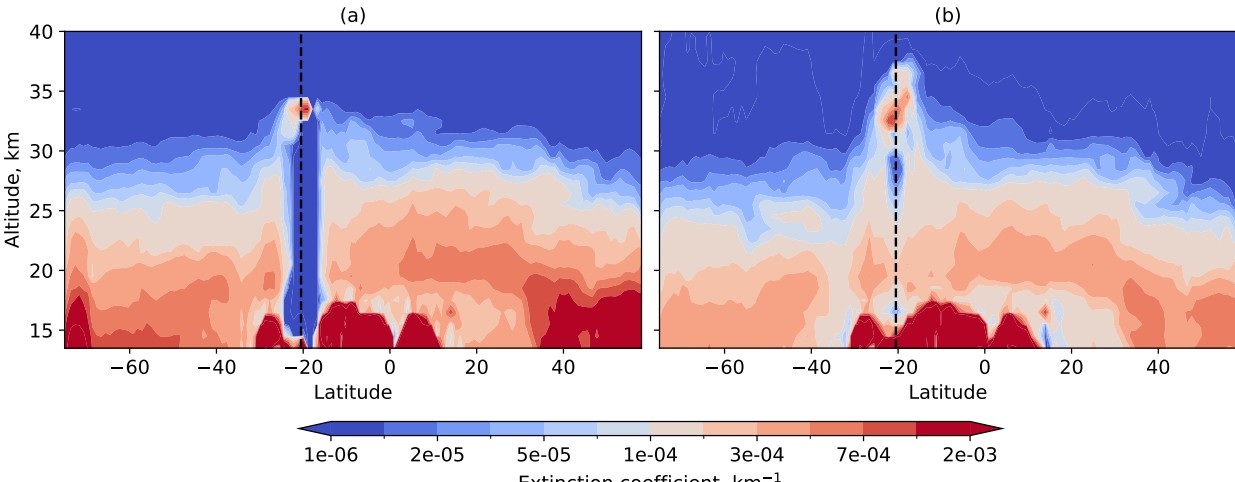

**Figure 4.** Aerosol extinction coefficient at 869 nm retrieved from OMPS-LP measurements during the orbit 52974 on January $17^{th}$, 2022. Panel (a): V1.0.9, which uses normalization to the reference tangent height. Panel (b): V2.1, which uses normalization to the solar irradiance. Dashed lines mark the location of the Hunga Tonga–Hunga Ha'apai volcano.

supplement. For a comparison, results obtained with V1.0.9 retrieval (but using the same Level 1 data as V2.1) are shown in Fig. S4 in the supplement. The values for the effective surface albedo retrieved for different test cases are shown in Table S1 in the supplement.

The dependence of the V1.0.9 retrieval, which uses the reference tangent hight normalization, on the aerosol amount around the tangent height of the reference measurement results in retrieval artifacts if a strong aerosol plume resides in this altitude
range. This effect is illustrated in Fig. 4, where retrieval results for an OMPS-LP single orbit shortly after the Hunga Tonga–Hunga Ha'apai eruption (January 2022) are presented. We note that here and below we use original V1.0.9 results rather than any adjustment of V2.1 to V1.0.9 settings. It is seen from the plot that V1.0.9 retrieval (panel (a)) shows unrealistically low values below the aerosol plume located in the reference tangent height region (38.5 km). As seen from panel (b) of the plot, this artifact is not present in the V2.1 retrieval, which uses the normalization to the solar irradiance. There is still a range of
quite small values below the aerosol plume seen in the panel (b) of the plot. For now, we cannot definitely say whether it is a real minimum in the aerosol extinction or a remaining retrieval artifact. It is also seen from the plot that V2.1 retrieval shows generally lower values than V1.0.9. As discussed below, these lower values are in a better agreement with reference measurements, which demonstrates another advantage of the new approach.

## 6   Comparison with other datasets

To assess the quality of the new retrieval and demonstrate the achieved improvement, the OMPS-LP results of versions 2.1 and 1.0.9 were compared to the colocated data from SAGE III/ISS. The criteria to select colocated pairs of measurements

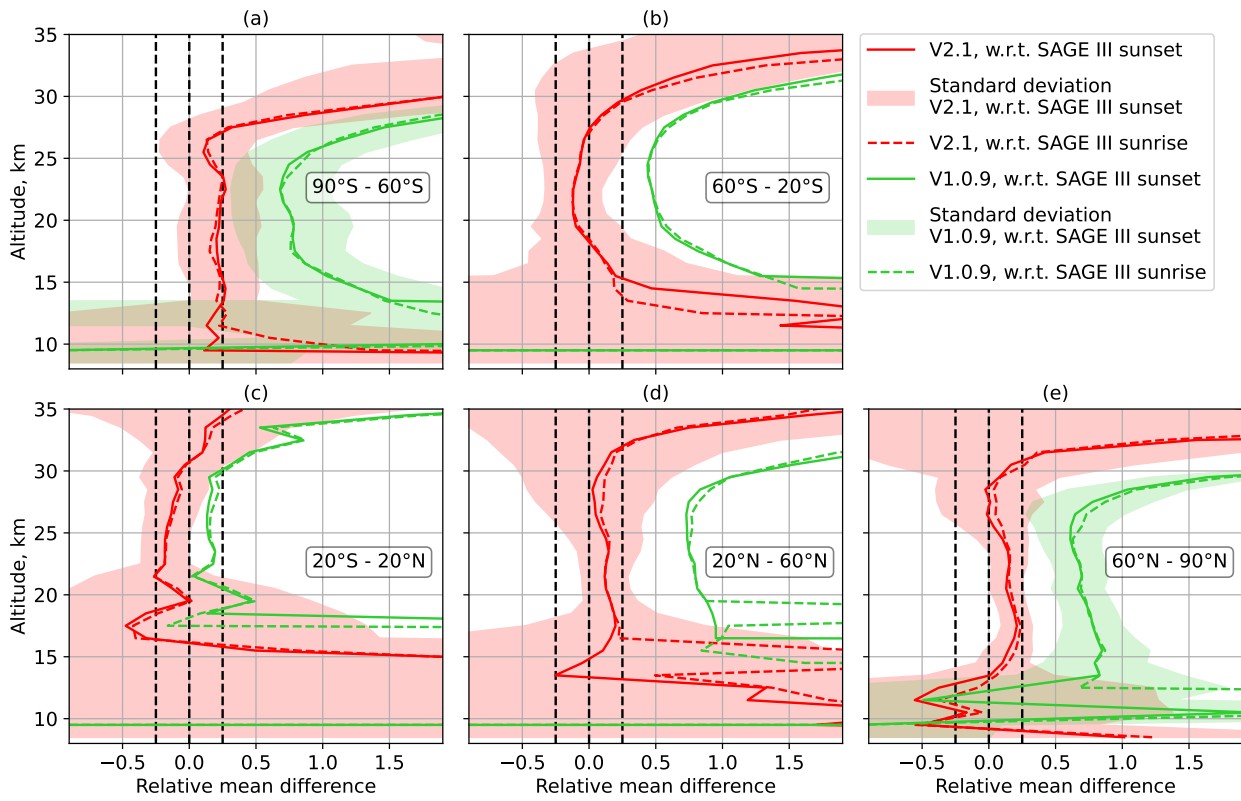

**Figure 5.** Comparison of V2.1 and V1.0.9 OMPS-LP retrieval results with colocated SAGE III/ISS data for 2018 for different latitude bands. The vertical black dashed lines mark 0% and ±25% levels.

were maximum differences of 5 degrees in latitude, of 10 degrees in longitude and of 12 hours. For this comparison no cloud filtering was applied for both data products. The comparison of the aerosol extinction coefficients at 869 nm from both instruments for the year 2018 (arbitrary selected) in different latitude bands is presented in Fig. 5. The relative difference was

calculated as (OMPS-LP - SAGE_III)/SAGE_III. The comparisons were done independently for SAGE III/ISS sunset (solid lines) and sunrise (dashed lines) measurements. For reasons of clarity, the standard deviation of the differences is shown only for colocations with the SAGE III/ISS sunset measurements. The plot demonstrates that V2.1 retrieval generally shows lower values than those from V1.0.9 significantly reducing differences to the SAGE III/ISS results for most of the latitude bands. In the tropics, the agreement with SAGE III/ISS data is similar for both V2.1 and V1.0.9 results with differences changing from

mostly positive for V1.0.9 to mostly negative for V2.1. Typically, V2.1 OMPS-LP retrieval agrees within 25% with SAGE III/ISS data between 18 and 30 km. Only in high southern latitudes the agreement starts degrading already above 27 km (see panel (a) of Fig. 5). In the lower stratosphere, a good agreement is seen down to about 13 km in the northern extra-tropics (about 16 km for sunrise colocations in the northern mid-latitudes), see panels (d) and (e) of Fig. 5, down to 15 km in the

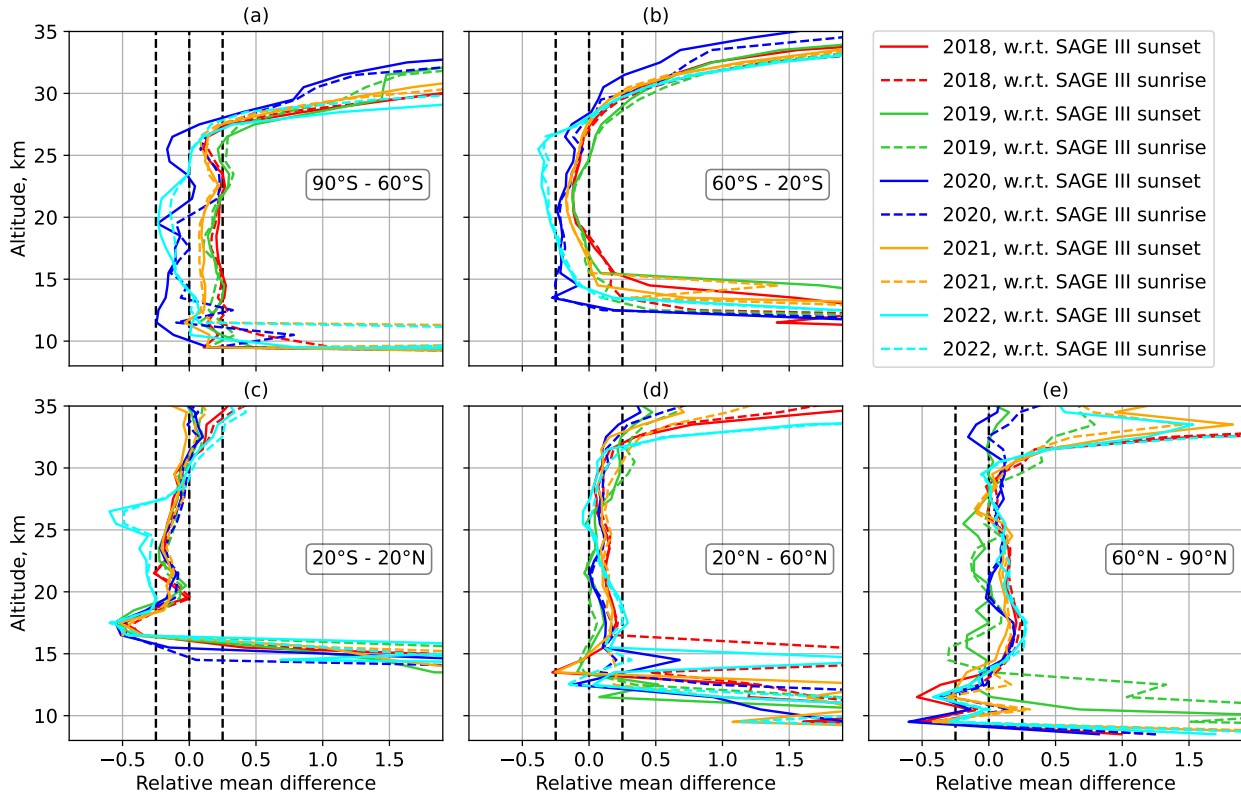

**Figure 6.** Comparison of V2.1 OMPS-LP retrieval results with colocated SAGE III/ISS data for different years and different latitude bands. The vertical black dashed lines mark 0% and ±25% levels.

southern mid-latitudes (panel (b)) and down to 8(12) km for sunset(sunrise) colocations in the southern high latitudes (panel
(a)).

Figure 6 presents relative mean differences between the aerosol extinction coefficients at 869 nm retrieved from OMPS-LP and SAGE III/ISS observations for different years. Only the V2.1 of OMPS-LP retrieval is shown. The plot reveals that differences between OMPS-LP and SAGE III data are very similar for most of the years and latitude bands. Exceptions are seen for (i) the tropics and southern extra-tropics in 2022 (panels (a) - (c)), where the influence of the Hunga Tonga–Hunga
Ha'apai eruption is the strongest, (ii) the southern extra-tropics (panels (a) and (b)) in 2020, which is most probably explained by the strong aerosol pollution from Australian wildfires, and (iii) the northern high latitudes (panel (e)) in 2019, most probably because of the Raikoke eruption. In general, the differences in the anomalous years are more negative indicating that OMPS-LP V2.1 aerosol extinction product is lower than SAGE III/ISS data in cases of a higher aerosol load in the stratosphere.

To evaluate a time evolution of the stratospheric aerosol extinction coefficients in more details, monthly zonal mean time
series from OMPS-LP V2.1 retrievals were compared with the results from SAGE III/ISS and OSIRIS instruments. A rough cloud filtering was done when calculating OMPS-LP V2.1 monthly zonal mean data by rejecting the extinction coefficient

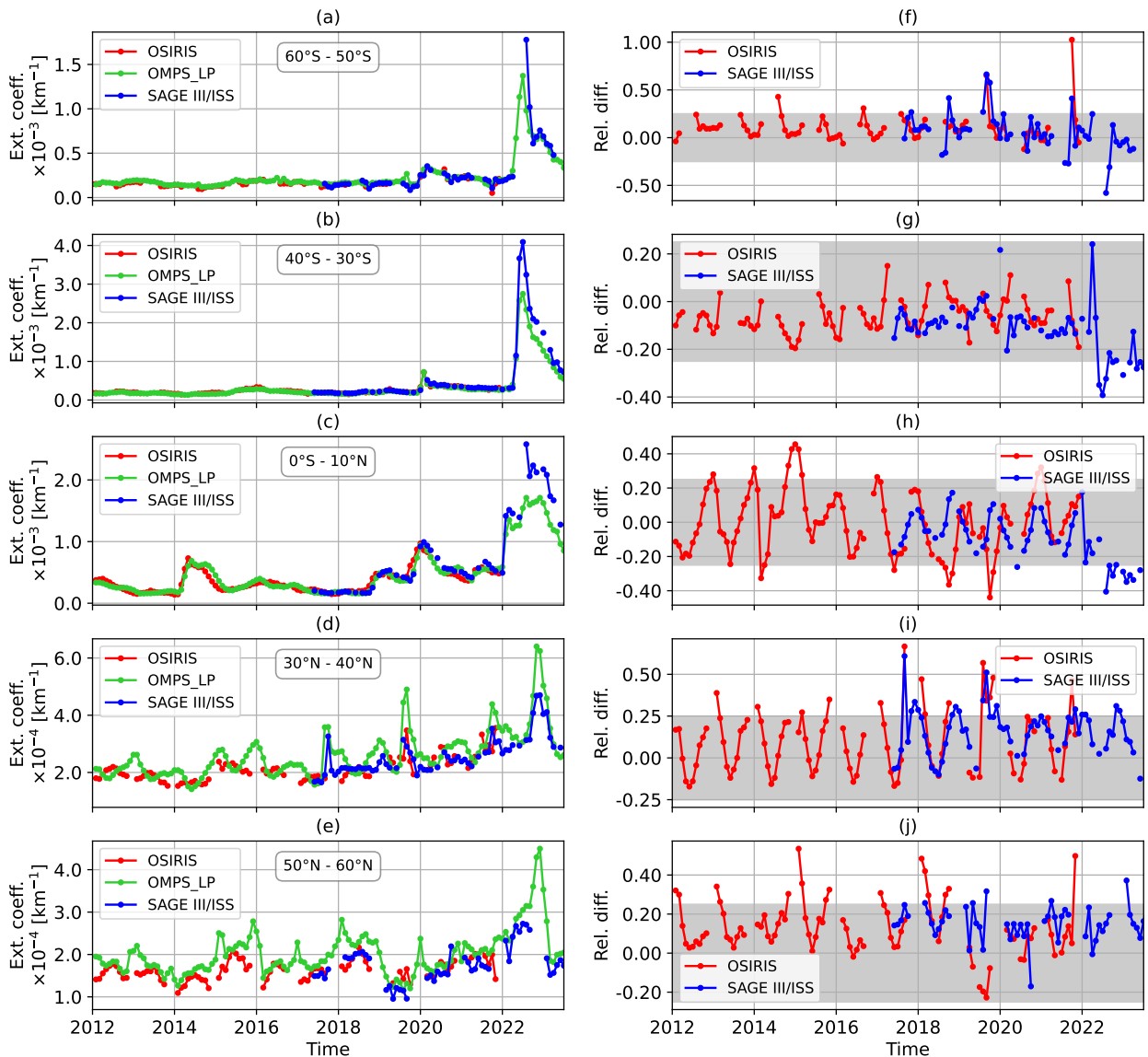

**Figure 7.** Comparison of monthly zonal mean aerosol extinction coefficients at 750 nm from OMPS-LP V2.1, OSIRIS and SAGE III/ISS data at an altitude of 21.5 km. Panels (a) – (e): aerosol extinction coefficient time series. Panel (f) – (j): relative differences. The rows show the results for different latitude bands: 60°S–50°S (panels (a) and (f)), 40°S–30°S (panels (b) and (g)), 0°N–10°N (panels (c) and (h)), 30°N–40°N (panels (d) and (i)) and 50°N–60°N (panels (e) and (j)). Grey areas mark ±25% range.

values larger than 0.1. To perform the comparison, the OMPS-LP data were converted to the wavelengths of 750 nm using the Ångström exponent calculated for 750 and 869 nm employing the Mie theory. The calculations were done assuming a fixed particle size distribution with the median radius of 0.08 $\mu$m and a standard deviation in the logarithmic space of 1.6 (same as

used in the retrieval, see Sect. 3). This results in a constant conversion factor of 1.477 when transforming data from 869 to 750 nm. For SAGE III/ISS the aerosol extinction coefficients at 756 nm were used without any adjustment. As mentioned in Sect.2, the University of Saskatchewan team does not recommend the usage of OSIRIS data after 2021. For this reason the comparison with OSIRIS is limited to the 2012–2021 period.

Comparison of the monthly zonal mean aerosol extinction coefficients at 750 nm from OMPS-LP (V2.1), OSIRIS and SAGE
III/ISS at an altitude of 21.5 km is shown in Fig. 7 for different latitude ranges. Panels (a) – (e) show the extinction coefficient time series while panels (f) – (j) depict relative differences between the respective time series and OMPS-LP data calculated as 2*(OMPS-LP - Instrument)/(OMPS-LP + Instrument). Generally, a good agreement between the results from different instruments is observed with relative differences ranging mostly within ±25%, which is consistent with the results presented in Figs. 5 and 6. After the Hunga Tonga–Hunga Ha'apai eruption, the agreement between the results from OMPS-LP and SAGE
III/ISS is worse in the 40°S–30°S (panels (b) and (g)) and 0°N–10°N (panels (c) and (h)) latitude bands. OMPS-LP values generally tend to be smaller than those from SAGE III/ISS. In the 50°N–60°N latitude band (panels (e) and (j)), OMPS-LP shows an increased aerosol loading in the beginning of 2023 which is not seen in SAGE III/ISS data. This might be caused by a coarser sampling of the latter instrument. In general, OMPS-LP results in this latitude band are slightly higher than those from SAGE III/ISS and OSIRIS. We note, that the OMPS-LP retrievals in this latitude band have the strongest sensitivity to
the aerosol particle size distribution because of a small scattering angle. In the 30°N–40°N (panels (d) and (i)) and 0°N–10°N (panels (c) and (h)) latitude bands, OMPS-LP data seem to show somewhat stronger seasonal cycle in comparison with both OSIRIS and SAGE III/ISS measurements and the pattern of the relative difference seems to be dominated by the seasonal cycle. The observed difference in the seasonal cycle is most probably caused by the dependence on the scattering angle of the retrieval error associated with an assumption of a fixed aerosol particle size distribution. As expected, for small scattering
angles as occurs at the middle and high northern latitudes for OMPS-LP observational geometry, this dependence is higher resulting in a pronounced seasonal variation of the differences between the results from OMPS-LP and those from SAGE III/ISS and OSIRIS. We note that OSIRIS instrument observes the atmosphere at a narrower range of scattering angles (60° – 120°) in comparison to that of OMPS-LP (25° – 160°).

To analyze the influence of seasonal variations, deseasonalized time series were investigated in addition. These were obtained
by calculating absolute anomalies of the time series, i.e. by subtracting from each monthly value the mean value for this month in the considered observation period, and then adding a mean value (its own for each instrument) over all months in the observational period. The comparison results for deseasonalized time series are presented in Fig. 8. The plot reveals that the differences are generally lower for the deseasonalized time series, especially in the 30°N–40°N latitude band (panels (d) and (i)). In contrast, the oscillating structure seen in the 0°N–10°N latitude band (panels (c) and (h)) before 2016 is not removed
by the deseasonalizing procedure, indicating that these differences are most probably not related to the seasonal variations. We note, however, that due to an orbit drift, the equator crossing time of OSIRIS changes with the time. Therefore, depending on the time and latitude, either the measurement at the ascending node of the orbit or that at the descending node of both of them are made in the illuminated part of the atmosphere and their results are included into the total data product. This might influence the efficiency of the deseasonalizing procedure.

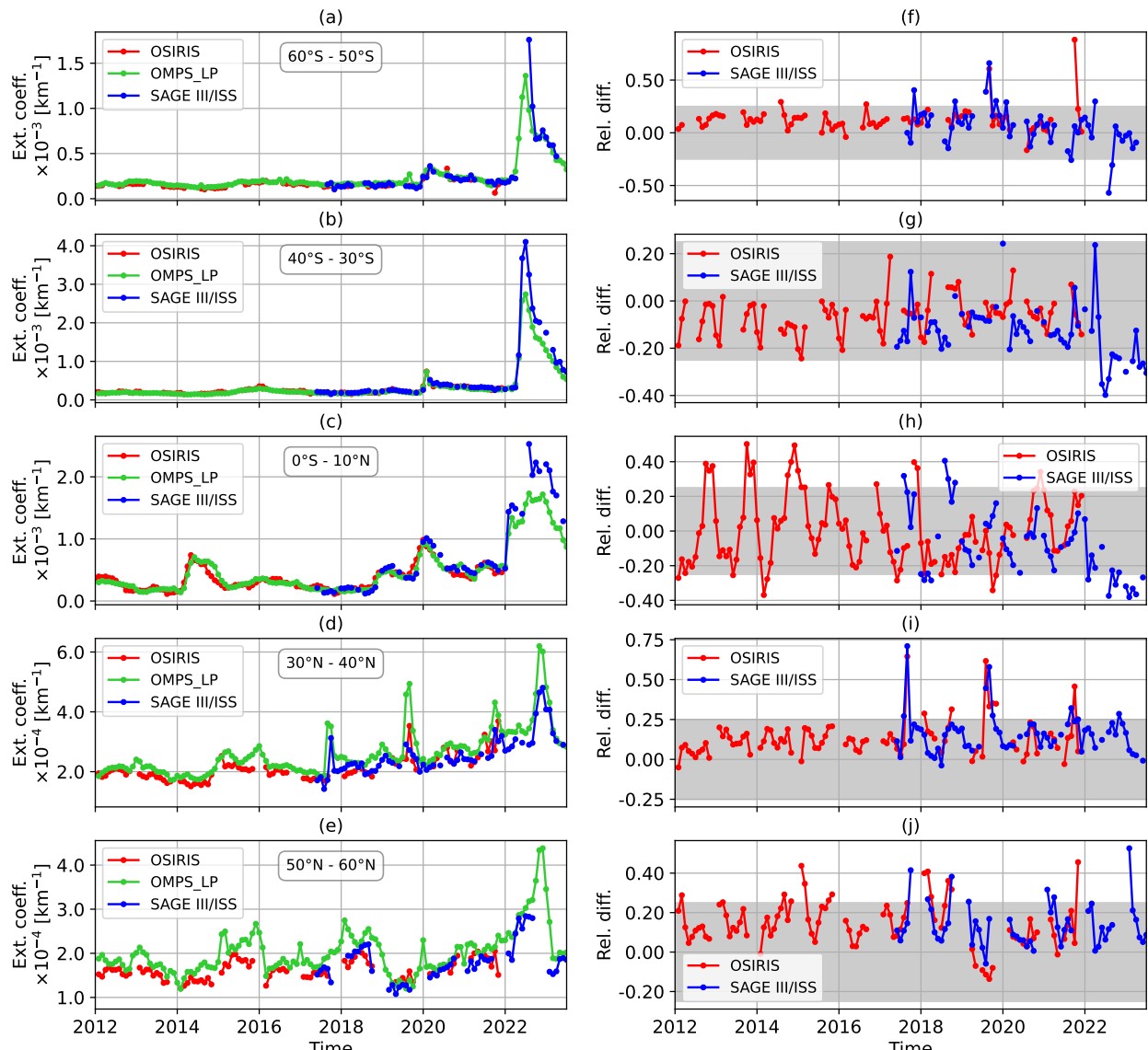

**Figure 8.** Same as Fig. 7 but for deseasonalized time series.

Figure 9 presents an overview of the relative differences between the OMPS-LP V2.1 and OSIRIS data as a function of the altitude and latitude. The differences were obtained by averaging all data in the 2012 – 2021 period. Below 60° latitude in the altitude range between the tropopause and about 30 km the differences between the results from both instruments are mostly within 10%. In contrast, higher differences are observed at latitudes above 60° below 20 km in the northern hemisphere and in the whole altitude range with the maximum around 30 km in the southern hemisphere. These differences might be related to the sampling issues of both instruments at high latitudes and to retrieval issues associated with a high surface albedo and

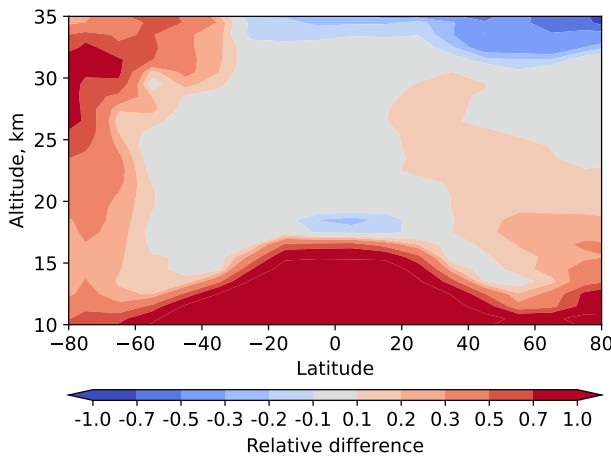

**Figure 9.** Comparison of zonal mean aerosol extinction coefficients at 750 nm from OMPS-LP V2.1 and OSIRIS data (2012 – 2021).

large solar zenith angles. As expected, high differences in the troposphere are observed because of a different treatment of the clouds.

Relative differences between the OMPS-LP V2.1 and OSIRIS data as a function of the altitude and time are shown in Fig. 10 for three different latitude ranges: southern mid-latitudes (60°S–20°S), tropics (20°S–20°N) and northern mid-latitudes (20°N–
60°N). Panels (a) – (c) of the plot show the results for original time series while panels (d) – (f) present the differences for deseasonalized time series. In the southern mid latitudes (panels (a) and (d)), both datasets agree typically within 10% in the altitude region between 20 and 27 km. Somewhat higher differences are seen in 2020, where the stratospheric aerosol loading was increased as a results of Australian wildfires. Above 27 km and below 20 km pronounced seasonal variations are seen in the relative differences with areas of higher differences grouping around the gaps in OSIRIS observations in the austral winter
380    months. Deseasonalizing the time series reduces the differences below 21 km to less then 20% but does not seem to have any effect above 27 km. In the tropics (panels (b) and (e)), the results from OMPS-LP and OSIRIS agree typically within 10-20% in the altitude region between 20 and 33 km. For the deseasonalized time series, the difference are slightly lower, however, the general picture remains almost the same. In the northern mid-latitudes (panels (c) and (f)) stronger seasonal oscillations are seen in the relative differences with areas of higher differences mostly grouped around the gaps in OSIRIS observations in
the boreal winter months. Somewhat higher differences are seen at the end of 2017 and the beginning of 2018, most probably associated with British Columbia forest fires. For the deseasonalized time series, the differences are largely smoothed and do not show any pronounced peaks with exception of the last months of the time series. Overall the OMPS-LP results are biased positively with respect to the OSIRIS data but the relative differences do not typically exceed 20% in the altitude range between 15 and 30 km.

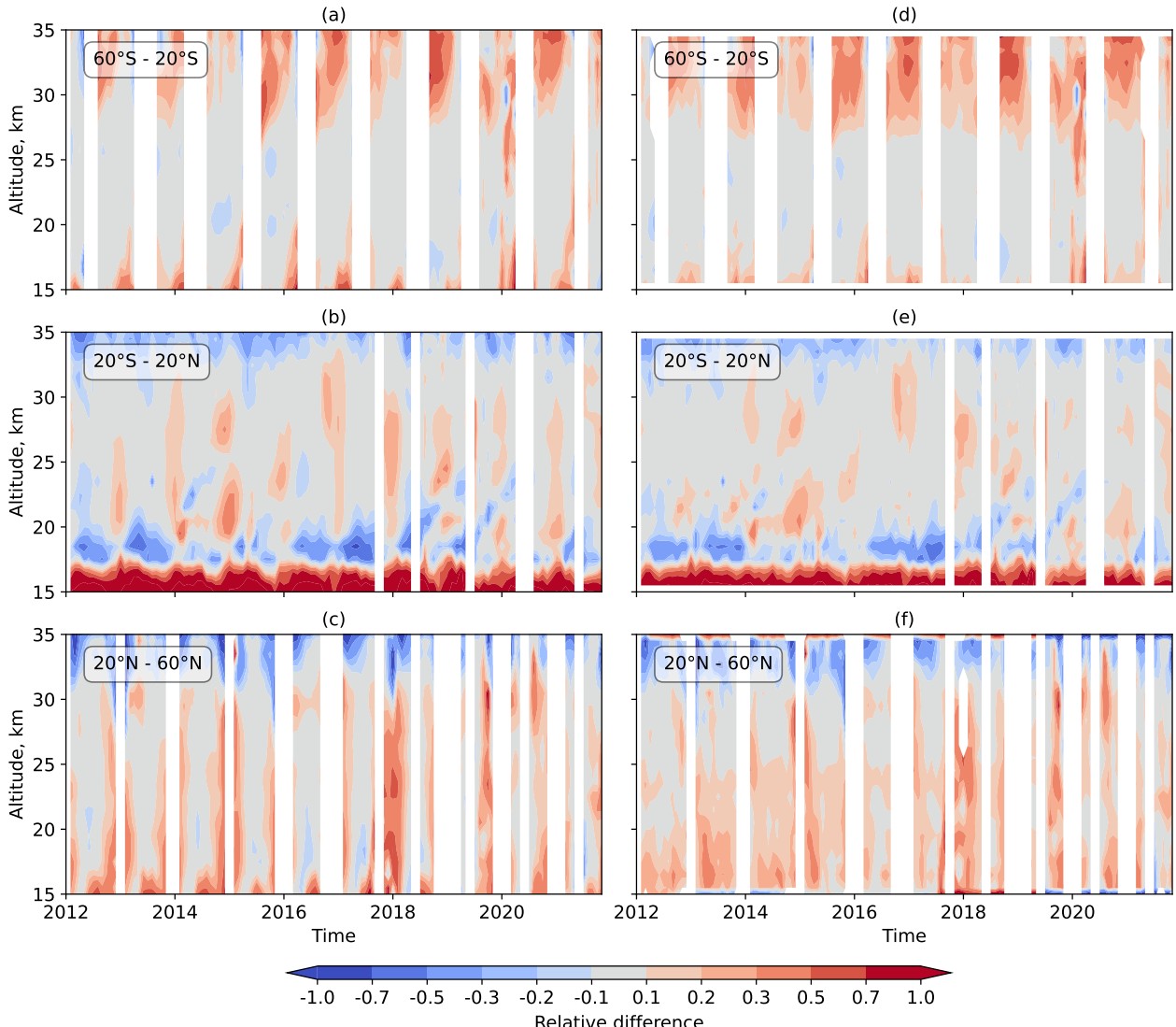

**Figure 10.** Relative differences of the monthly zonal mean aerosol extinction coefficients at 750 nm from OMPS-LP V2.1 and OSIRIS data for different latitude ranges: 60°S–20°S (panels (a) and (d)), 20°S–20°N (panels (b) and (e)), 20°N–60°N (panels (c) and (f)). Panels (a) – (c): aerosol extinction coefficient time series. Panels (d) – (f): deseasonalized time series.

## 7   Evolution of the aerosol plume from the Hunga Tonga–Hunga Ha'apai eruption

In this section the newly generated OMPS-LP V2.1 aerosol extinction coefficient product is used to investigate the evolution of the aerosol plume after the Hunga Tonga–Hunga Ha'apai eruption in January 2022. For this purpose a dataset of zonal mean aerosol extinction coefficients at 869 nm with an increased temporal resolution (10 days mean) was created. The altitude-time cross sections of the aerosol extinction coefficient for different latitude ranges are shown in panels (a) – (c) of Fig. 11. In

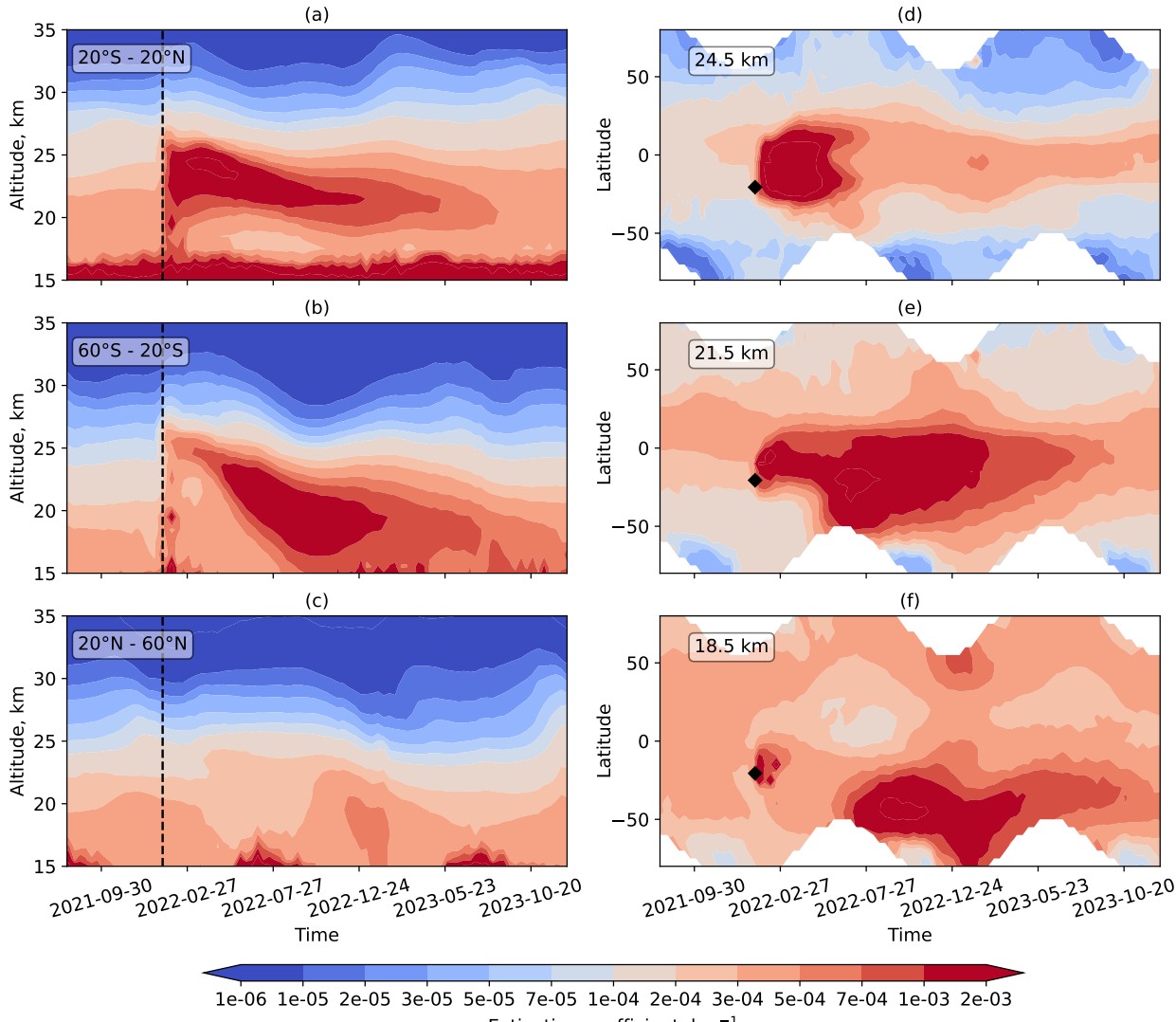

**Figure 11.** Evolution of the aerosol plume in different latitude bands and at different altitude levels. Panels (a) – (c): 10 days zonal mean aerosol extinction coefficient at 869 nm in different latitude bands: 20°S–20°N (panel (a)), 60°S–20°S (panel (b)), 20°N–60°N (panel (c)). The black dashed line mark the time of the strongest Hunga–Tonga eruption (January 15$^{th}$, 2022). Panels (d) – (f): 10 days zonal mean aerosol extinction coefficient at 869 nm at different altitude layers: 24.5 (panel (d)), 21.5 (panel (e)), 18.5 km (panel (f)). The black rhombus marks the position of the Hunga Tonga–Hunga Ha'apai volcano and date of its strongest eruption. The OMPS-LP data of V2.1 were used for all plots.

the tropics (panel (a)) the aerosol extinction coefficient raises right after the strongest eruption of the Hunga Tonga–Hunga Ha'apai occurred on January 15$^{th}$, 2022 (marked by the vertical dashed line in the plots). A significant perturbation in the

vertical distribution of the extinction coefficient is seen up to about 33 km. It should be noted here, that the extinction observed right after the eruption is strongly affected by the ash plume and ice clouds, which might result in high retrieval errors caused by a wrong assumption about the aerosol composition in the Mie scattering calculations of the forward model. In about one time step, i.e. 10 days, a strong increase in the aerosol extinction coefficient is observed, which is most probably caused by a creation of a significant amount of the sulfate aerosols from the sulfur injected into the stratosphere by the volcanic eruption. As reported e.g. by Legras et al. (2022), ash and ice clouds were rapidly removed within the first day after the eruption while the conversion of $SO_2$ to sulfates started immediately after the eruption. Thus we can assume that from the second time bin after the eruption the extinction by sulfate aerosols dominates and the retrieval results are robust. It is seen from the plot that the maximum of the extinction coefficient is observed between about 21 and 26 km. The plume remains stable until about the middle of April 2022 and then starts sinking. Above about 22 km the values quite rapidly relax to quasi-stationary levels, which are significantly higher than those before the eruption.

In the southern mid-latitudes, as shown in panel (b) of the figure, a similar immediate increase of the aerosol extinction coefficient after the Hunga Tonga–Hunga Ha'apai eruption as in tropics is observed. However, rather few sulfate aerosols are observed 10-20 days after the eruption. A stronger increase in the aerosol extinction coefficient is seen only after the middle of April, when the aerosol plume in tropics started sinking and was transported into the southern mid-latitudes. Thereafter, the evolution of the plume was quite similar to that in the tropics. The plume was sinking and the values near the upper boundary of the plume quite rapidly relaxed to quasi-stationary elevated values. In the northern mid-latitudes, as shown in the bottom left panel, no instantaneous response to the Hunga Tonga–Hunga Ha'apai eruption is observed. Elevated values are observed at lower altitudes since November 2022, which are most probably caused by the transport of the Hunga Tonga–Hunga Ha'apai aerosols from the tropics. As discussed below, this statement is confirmed by the evolution of the aerosol plume observed in the latitude-time cross section plot at 21.5 km (panel (e)).

In panels (d) – (f) of Fig. 11, the latitude-time cross sections of the aerosol extinction coefficient for different altitudes are shown. The latitude of the Hunga Tonga–Hunga Ha'apai volcano and the date of its strongest eruption are marked by the black rhombus. At 24.5 km (panel (d)), large amounts of the aerosols were created about 10 days after the eruption. The strongly elevated aerosol plume formed over the entire tropical region and partially penetrated to the southern sub-tropics. In the first four months after the eruption, the plume did not extend much in the latitudinal direction and resided mainly in the tropical region. Thereafter, a propagation of the aerosol plume to the southern mid-latitudes is observed. From June 2022, the aerosol extinction coefficient started rapidly to decrease reaching a quasi-stationary elevated level at the end of July. In spring 2023, the plume starts disappearing in the southern sub-tropics and mid-latitudes but remains stable until the end of the analyzed record (December 2023) in the tropics.

At 21.5 km (panel (e)), the aerosol plume was localized in tropics within the first three months after the eruption. In May 2022, the plume rapidly propagated southwards and spread over the entire southern mid-latitudes. No propagation to the northern extratropics is seen until October 2022. Thereafter, the plume spread to the northern hemisphere.

At 18.5 km (panel (f)), a strong increase of the extinction is seen right after the eruption. Elevated values can only be observed until the end of March, before the concentration fell back to the pre-eruption level. From the beginning of June 2022, the high

aerosol amounts, which propagated to the southern mid-latitudes from tropic at around 21.5 km altitude, sank to 18.5 km leading to a strong increase of the aerosol extinction in the almost entire southern hemisphere. These increased values resided at this altitude until May 2023. Thereafter a relaxation process is observed. At the end of the analyzed period (December 2023)

some increase in the aerosol extinction coefficient is still observed. In the northern hemisphere, increased values of the aerosol extinction coefficient are observed at the end of 2022 and the beginning of 2023. It is, however, unclear whether these values are related to the Hunga Tonga–Hunga Ha'apai eruption plume.

The results presented in this section are in a good agreement with the findings of other studies on the evolution of the stratospheric aerosol after the Hunga Tonga–Hunga Ha'apai volcanic eruption. For example, using a ground-based lidar, Baron

et al. (2023) observed a strong aerosol plume with a maximum just below 30 km on January $21^{st}$, 2022 (5 days after the eruption) over Reunion island (21°S, 55°E). Within the next few days, aerosol plumes with the altitude decreasing to about 20 km were observed. This agrees with an aerosol extinction peak at altitudes up to about 30 km and an increase in aerosols over the entire altitude range shortly after the eruption seen in panels (a) and (b) of Fig. 11. Analyzing SAGE III/ISS data, Duchamp et al. (2023) report a strong increase of the aerosol extinction shortly after the eruption in 30°S – 10°S and 10°S

445 – 10°N latitude ranges with peak altitudes between 20 and 26 km. The maximum altitude of the plume remains stable until May-June 2022 then the plume starts descending. These findings are in agreement with the results presented in panel (a) of Fig. 11. At southern mid-latitudes (50°S – 30°S), Duchamp et al. (2023) report a strong increase of the aerosol extinction starting from about May 2022 with the peak altitude descending from about 24 to 18 km and an onset of the relaxation after January 2023. A very similar behavior is observed in our Fig. 11 (panel (b)). In accordance with the plots presented by Taha

et al. (2022), who analyzed the OMPS-LP data (NASA retrieval), in the first 5 months after the eruption the aerosol plume in tropics resided between about 19 and 26 km altitude. No significant changes in its vertical extent and maximum altitude were observed. This agrees very well with our findings in panel (a) of Fig. 11. Furthermore, Taha et al. (2022) show that in the first 5 months the plume was always to the south of 20°N and started propagating southwards of 30°S after about 4 months after the eruption. This behavior also agrees well with our findings in panel (d) – (f) of Fig. 11.

**8   Conclusions**

In this study we have introduced a new retrieval algorithm developed to obtain vertical distributions of the aerosol extinction coefficient from limb-scatter measurements of the OMPS-LP instrument. The main change with respect to the algorithms used by other scientific groups as well as with respect to the previous version of the University of Bremen retrieval is the normalization of the limb measurements by the solar irradince rather than by a limb measurements at an upper tangent height.

As normalizing to the solar irradiance increases the sensitivity of the retrieval to the reflectance of the underlying scene, an optimized scheme to derive the effective Lambertian surface albedo has been implemented.

The employed normalization approach makes the retrieval results from OMPS-LP V2.1 algorithm almost independent on the prior profile of the aerosol extinction coefficient used in the retrieval. In contrast, an adequate knowledge of the aerosol extinction at altitudes around the normalization tangent height is crucial for algorithms using the normalization to a limb

measurement at an upper tangent height. Furthermore, as the quality of limb measurements decreases with increasing tangent height and the influence of the stray light increases, skipping the normalization to an upper tangent height enables us to extend the vertical range of the retrieval to 8.5 - 48 km in comparison with 12.5 - 37.5 km retrieval range of the previous version of the University of Bremen retrieval, published by Malinina et al. (2021). This makes the retrieval applicable to the scenes with high aerosol plume elevation, as it was the case after the Hunga Tonga–Hunga Ha'apai eruption in January 2022.

The new dataset of the OMPS-LP aerosol extinction coefficient V2.1 covering the period from February 2012 to December 2023 was verified using the data from the solar occultation SAGE III/ISS and limb-scatter OSIRIS instruments. Comparison of the newly generated OMPS-LP V2.1 data product with that of SAGE III/ISS shows a good overall agreement within typically 25% between 15 and 30 km. Similarly, the comparison with OSIRIS data generally shows a good agreement of the temporal behavior and values. Within strong aerosol plumes, larger disagreement between the data from different instruments (OMPS-475 LP, OSIRIS, SAGE III/ISS) were identified. This is most probably related to the assumption about the aerosol particle size distribution used in the limb-scatter retrievals (OMPS-LP and OSIRIS), which was originally selected for background aerosol conditions and might be sub-optimal for strong volcanic eruption conditions.

The OMPS-LP V2.1 aerosol extinction coefficient dataset was used to investigate the evolution of the aerosol plume after the Hunga Tonga–Hunga Ha'apai eruption. The perturbation of the aerosol layer is seen up to the altitude of about 33 km. A 480 strong increase in the aerosol loading is seen up to the altitude of 27 km. About two weeks after the eruption the maximum of the aerosol absorption is seen in tropics between 21 and 26 km altitude. About 4 months after the eruption, the aerosol plume started rapidly sinking. At higher altitudes (23 -25 km) the plume mainly resided in tropics, while at lower altitudes (around 18 km) the aerosol was transported southwards and spread over almost the entire southern hemisphere. At higher altitudes the relaxation already started, but at the end of the analyzed period (December 2023) the aerosol level in tropics still remained 485 elevated in comparison to the pre-eruption level.

The OMPS-LP V2.1 aerosol extinction coefficient dataset described in this manuscript is available for the scientific community via the web page of the University of Bremen (see data availability section below).

*Data availability.* The OMPS-LP V2.1 aerosol extinction coefficient of IUP Bremen is available at https://www.iup.uni-bremen.de/DataRequest

*Author contributions.* AR developed the OMPS-LP V2.1 retrieval algorithm, generated the data set, evaluated the results and written the 490 initial text of the manuscript. CP participated in the algorithm developments and made initial evaluation. CA prepared tools for an initial processing of OMPS-LP Level-1 data, KB provided access to all necessary Level-1 and Level-2 data. AB and LR provided Level-2 OSIRIS data and support on their usage. EM developed OMPS-LP V1.0.9 data set and provided the data for the comparison. JB did an overall scientific supervision of the study. All authors contributed to the discussion and improvement of the initial manuscript.

*Competing interests.* The authors declare that they have no conflict of interest.

*Acknowledgements.* This study was funded in parts by the European Space Agency (ESA) via CREST project, by the German Research Foundation (DFG) via the Research Unit VolImpact (grant no. FOR2820), and by the University and State of Bremen. The authors gratefully acknowledge the computing time granted by the Resource Allocation Board and provided on the supercomputer Lise and Emmy at NHR@ZIB and NHR@Göttingen as part of the NHR infrastructure. The calculations for this research were conducted with computing resources under the project hbk00098.

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
