# Peer review of "Retrieval of stratospheric aerosol extinction coefficients from OMPS-LP measurements"

_EGUsphere, 2024_

## Referee Comment (RC2)

Review of 2024-358, Rozanov

General:

The authors present a new retrieval scheme for aerosol extinction coefficient derived from profiles of limb-scatter radiance as applied specifically to OMPS-LP data. Advantages over their previous algorithm are shown in a convincing manner. While there are several differences between the two algorithm versions, the authors state the overruling factor is the normalization approach. The manuscript would benefit from a sensitivity analysis for typical error sources, even a limited one would be enlightening. Highlighting results from the Hunga eruption is very nice. I recommend publishing this article, after successfully addressing the comments below.

Specific:

Is there a missing affiliation in the list on the cover page, i.e. 3?

Lines 136-144: Need to include quantitative estimates of the retrieval errors.

Section 4: Should make it clear that there are only a finite number of wavelengths to use since OMPS-LP does not download the full spectrum. That would be why you are not suggest using slightly different wavelengths to reduce interference from atmospheric absorption/emission.

Line 199: Should be $O_2$ – B instead of $O_2$-A and wavelength is 688nm (Newnham & Ballard, 10.1029/98JD02799) (Visible absorption cross sections and integrated absorption intensities of molecular oxygen (O2 and O4) - Newnham - 1998 - Journal of Geophysical Research: Atmospheres - Wiley Online Library)

Line 200: Should be $O_2$-A instead of $O_2$-B

Line 204: $O_2$ has a band near 867nm

Figure 3: What solar irradiance spectra are used for these cases?

Figure 3: What do the aerosol profiles look like with the new algorithm for altitudes above 35km? This should help the OMPS team know how well the stray light correction scheme performs.

Line 244: The normalization range does seem to be too low in altitude. If the solar normalized radiances up to 50 km are good enough to estimate surface albedo, then the normalization range for the V1.0.9 retrieval should be raised, maybe above 45 km.

Line 253: What are the reflectance values for the various cases?

Line 254: "scaling of the a priori at the…reference tangent height" This appears to behave differently than trace gas retrievals from UV/VIS backscattered sunlight that use a normalizing spectra obtained from backscattered radiance spectra over a reference sector, typically a region with low trace gas amounts. The trace gas amounts from the reference sector are subtracted from the total trace gas.

Line 332: Is the same mean added to all three datasets to make the left panel of Fig. 8?

---

## Editor Comment (EC1)

Review of "Retrieval of stratospheric aerosol extinction coefficients from OMPS-LP measurements" by Rozanov et al. (2024) to be published in AMT

The stratospheric aerosol layer plays a critical role in the Earth's climate system through its impact on radiation, chemistry and the hydrological cycle. Impacted by large volcanic eruptions, its composition and loading can also reflect the influences of sulfur precursor emissions ($SO_2$, OCS, DMS), extreme wildfires through Pyro-convection and the Asian Summer Monsoon transport pathways. The stratospheric aerosol layer has been studied since more than 4 decades through satellite-based solar occultation techniques, ground-based lidar and balloon-borne observations. More recently, limb observations have shown its ability to study stratospheric aerosol despite some limitations on calibration procedures, resolving complex radiation influence from scattering and absorbing and underlying assumptions on aerosol size distribution. Rozanov et al. (2024) utilizes the Ozone Mapping Profiler Suite- Limb Scatter instrument to study stratospheric aerosol extinction quasi-globally since 2012. Improvements of the retrieval algorithms are discussed in this paper and the results are compared with SAGE III/ISS and OSIRIS satellite observations. Overall, the strengths and limitations of the new algorithm are well exposed and convincing. This is a well-written, logically-structured and organized paper which merits to be published in AMT after some minor corrections can be applied and additional explanations could be provided.

1) L15P1: Solomon et al. 2011 do not report the presence of large amount of aerosols but rather an increase of stratospheric aerosols from moderate but frequent volcanic eruptions as reported by Vernier et al. (2011). I would recommend correcting this sentence.

2) L25-26P2: Evan et al. 2023 report ozone loss soon after the HTHH eruption with limited explanations about the causes. Zhu et al. (2023) found that enhanced chlorine from marine sources was likely responsible of the ozone loss more than a week after the eruption rather than dynamical processes. The same study evokes a different ozone loss mechanism than traditional volcanic eruptions. I believe that some nuances could be made here.

3) P2L29: This statement should be nuanced and is not fully correct. SAGE has provided quasi-global observations since 1979 but at rather low spatial sampling (30 profiles per day)

4) P2L35: While describing SAGE data, some information regarding the fact that the spectra are self-calibrated through exo-atmospheric measurements might be of interest for the reader.

5) P3L62: I do not believe that this paragraph justifies well why CALIPSO is not used. As a matter of fact, I would recommend using the new stratospheric aerosol product level 3 developed recently (asdc....). It could be used to understand the performance of OMPS algorithm when other datasets are not available (e.g. SAGE III/ISS in the polar winter regions Or near the tropopause where the variability of aerosol might be important and the influence of cirrus clouds in the tropics significant). In addition, I could not find how OMPS and other measurements were collocated with OSIRIS and SAGE III/ISS. Could you please clarify this >

6) P4L107: This is extremely difficult to make sense of this for non-specialist. I recommend to use some references but also to provide additional information by trying to avoid employing too many technical terms. Maybe a schematic describing the different steps of the

algorithm could be useful here. Additional effort should be made here to further explain the different steps of the algorithm.

---

## Author Comment (AC1)

We thank the reviewer for carefully reading our manuscript and helpful comments. Below, the reviewer's comments are marked in blue and our answers to the comments are written in black.

The manuscript "Retrieval of stratospheric aerosol extinction coefficients from OMPS-LP measurements" by Rozanov et al. presents a new retrieval algorithm to obtain vertical profiles of the aerosol extinction coefficient. The main claim of the paper is that by avoiding altitude normalization, the algorithm becomes almost completely independent of the "prior aerosol extinction profile." However, in my view, the authors did not provide sufficient evidence to prove this point. Particularly, in Section 5, the authors wrote that uncertainties about the aerosol concentration at the normalization altitude would lead to a strong sensitivity to the a priori extinction profile across the entire vertical range. Figure 3 shows results for two algorithms (V1.0.9 and V2.1) and concludes that by removing altitude normalization in V2.1, the retrieved profiles become almost insensitive to a priori. However, I am afraid that the authors are comparing apples and oranges here. We (readers) do not know if the two algorithms use the same L1 data or different data because the differences in the magnitude of retrieved aerosol extinction coefficients are quite large between the two algorithms, as shown in Figure 5. The authors do not describe all the algorithmic differences between the two algorithms to convince the reader that the changes they see in Figure 3 are caused by the normalization at higher altitude.

Actually, the investigation was done in exactly the same way as suggested by the reviewer. The version we denoted in this part of the manuscript as V1.0.9 was actually V2.1 with the normalization switched to the upper tangent height (same as in V1.0.9). We thought using the notation V1.0.9 makes the things more easy for the reader, this seems, however, to cause confusion. We apologize for this. In the revised manuscript we changed the notations and revised the corresponding text.

The authors listed three main reasons why altitude normalization can negatively affect aerosol retrievals: larger stray light at the normalization altitude, uncertainties about the aerosol amount at the normalization altitude (that comes from a priori), and scene reflectivity (albedo). I agree with all three points; however, I don't understand how any of these factors can lead to a strong dependence on the a priori throughout the entire vertical range. To prove the claim, the authors perturbed a priori profiles by increasing them by a factor of 2 and 3 and ran retrievals using the two models. First of all, if the authors believe that it's the uncertainties in aerosol concentration at the normalization altitude that affect aerosol retrievals below, then they should arbitrarily increase aerosol at the normalization altitude rather than the entire profile.

We prefer to keep the plot with scaling the entire a priori profile in the main text of the manuscript for the reason that this makes our point easier to understand (changing the prior profiles only above the reference tangent height does not change them within the plot range). To address the reviewer's comment

we added a plot to the supplement showing the changes in the resulting profiles when we change the a priori profile only at and above the reference tangent height and only below the reference tangent height, respectively.

The retrieval sensitivity to a priori can also be estimated using AKs (see Rodgers, 2000). The AKs for V2.1 are shown in Figure 2, but V1.0.9 AKs had not been demonstrated to readers. Can you please plot them as well? Can you estimate sensitivity to a priori using the equation (Rodgers et al., 2000) and check if it's consistent with what you observe from direct perturbations of the a priori?

The averaging kernels for V1.0.9 look very similar to those from V2.1. The reason for that is that both retrievals are almost independent of the a priori information in the retrieval range, i.e. below the reference tangent height. The fact that V1.0.9 retrieval is strongly sensitive to the a priori profile at and above the reference tangent height, i.e. outside the retrieval range, is not reflected in the averaging kernels, as it is not a dependence on the a priori information in the sense of Rodgers (2000). Following the terminology of Rodgers (2000), it is rather a parameter error. For this reason, the formalism for the estimation of the influence of the a priori information described by Rodgers (2000) is not useful in this case.

Secondly, the authors claim that scene albedo R derived at 40 km depends on aerosol, which is true, but I am not sure how that can increase the sensitivity to the a priori aerosol. Since the background aerosol amount is negligibly small at 40.5 km, its contribution to R is quite small compared to the pure Rayleigh atmosphere. I agree that the change in R will affect aerosol retrievals, but I am not sure how that can increase the sensitivity to the a priori aerosol. The authors should provide evidence to support this claim.

We would like to point out that the reference tangent height in V1.0.9 was at 37.5 km rather than at 40.5 km. At least at this altitude the aerosol signal is not negligibly small. If this was the case, there would be no dependence on the a priori profile used at these altitudes (see Fig. S1 in the new supplement). As the assumed aerosol around 37.5 km has a non negligible contribution to the measured signal it is also non-negligible when retrieving the effective albedo. The smaller our assumed aerosol concentration at 37.5 km, the larger the retrived effective albedo becomes. This is because it compensates for the signal missing due to the underestimation of the aerosols. A bias in the retrieved albedo produces, in turn, an additional retrieval error. We agree with the reviewer that this effect is small for very clean upper stratospheric conditions but this is certainly not the case if a strongly elevated aerosol plume is observed as e.g. for the Hunga-Tonga eruption (see Fig. 4 of the paper).

I believe the method proposed in this paper can be useful for aerosol retrievals in perturbed conditions like those after the Hunga eruption, when the aerosol at  40 km was significantly different from the climatology. However, in

my view, the paper needs to be substantially revised, and the authors have to provide more supporting evidence for their main claim.

As we discussed above, we consider that the concerns of the reviewer were a result of our not explaining adequately the dependence on the a priori profile shown in Fig. 3 of the manuscript. Figure 3 shows the difference between V2.1 running similarly to V 1.0.9 and V2.1. This difference is determined by the aerosol concentration at and above the reference tangent height. We hope that our clarification about these differences between the retrievals and new Fig. S1 in the supplement sufficiently address the issue raised by the reviewer.

Major Comments: Title: The title of the paper is too vague and does not reflect the content of the paper. As the authors pointed out, there are multiple groups and multiple aerosol retrieval algorithms that use OMPS LP measurements to derive aerosol extinction. The title should be changed to reflect the paper's content.

We added "sun-normalized" to the title to make it reflect more clearly the objectives and ambition of our manuscript.

Abstract, line 4: There is a statement in the paper, which is repeated multiple times, saying that the novelty of the presented algorithm is that "the method employs the normalization of the limb radiances to the solar irradiance in contrast to the normalization by a limb measurement at an upper tangent height, which is used by most of the other published limb-scatter retrievals." A search in the literature reveals that, for example, NASA's retrieval algorithm (Loughman et al., 2018; Chen et al., 2018; Taha et al., 2021) uses sun-normalized radiances to derive aerosol extinction. Indeed, the NASA algorithm requires the altitude normalization at higher altitudes, but it is incorrect to state that nobody uses sun-normalization.

We disagree with the reviewer. Loughman et al. (2018) use the altitude-normalized radiances (ANRs) for their retrieval, which is the radiance at a tangent height of interest divided by the radiance at a selected normalization tangent height (see the beginning of their section 3.4). Chen et al. (2018) write at the beginning of their Sect. 2 " the radiances are normalized (i.e., divided by their value at the normalization altitude, 40.5 km) in all cases", Taha et al. (2021) write in their section 2.2.2: "To reduce the stray light effect on the retrieval at longer wavelengths, $h_n$ was lowered to 38.5 km in V2.0 (from the 40.5 km value used in previous versions)." In summary, all these manuscripts are reporting the results of algorithms which use the reference tangent height normalization. They certainly have used the sun-normalized radiances before the additional normalization as it is provided in Level-1 OMPS-LP product, this has, however, no meaning with respect to the discussed topic as the solar normalization cancels out when dividing by the reference tangent height. We reworded the sentence in the abstract to clarify that the solar irradiance normalization means the absence of the normalization to the reference tangent height.

Abstract: A large fraction of the paper is dedicated to comparisons with other instruments (SAGE III and OSIRIS). The statement in the abstract declares that differences are mostly within 25%, but such agreement is only seen in a relatively narrow vertical range, and outside that range, the differences are much larger. In my view, the authors should clearly identify in the abstract the vertical and latitudinal ranges where the agreement is within the desired 25%.

We included the vertical range for both comparisons and the latitudinal limitation for OSIRIS in the abstract

Page 2, lines 24-26: The authors stated that substantial ozone losses were observed after the 2020 Australian fires and the Hunga eruption and provided references. In my view, the words "significant losses" exaggerate the losses described in the cited studies. Instead of using the words "significant ozone losses," the authors should quote numbers from the cited publications.

We added the numbers from the papers as recommended by the reviewer.

Page 3, line 57: There is an extensive list of publications that estimate the SO2 amount injected by the Hunga eruption. It would be better to quote numbers rather than say "a significant amount."

We quoted numbers in accordance with the reviewer's recommendation an added additional references to support these numbers.

Page 4, Section 3, line 115: Are you solving Equation 1 with respect to the initial guess or a priori profile? Is the first guess in your terminology the same as a priori?

Yes, initial guess profile is the a priori profile. This is clarified in the revised manuscript.

Page 4, line 116: By removing the altitude normalization, you need to accurately know the surface albedo. Can you reduce the number of iterations by retrieving reflectivity R0 at, say, 40 or 45 km first and use this as the initial guess for R?

In the standard retrieval, the effective surface albedo is quite close to the final value already after 2-3 iterations. Thus, there is no advantage in doing a pre-retrieval, especially taking into account that the effective albedo retrieved from measurements at upper tangent heights might be wrong because of a correlation with the stratospheric aerosol signal or stray light contamination.

Page 5, Section 3, lines 26-28: It is not the normalization to solar radiances that makes retrievals more sensitive to upwelling radiances. It is the absence of the altitude normalization.

Changed to "As the normalization by the solar irradiance instead of the reference tangent height makes the retrievals more sensitive to the surface reflectance ..." to avoid a confusion.

 The described convergence criteria are questionable and definitely are not optimal. The range between 15 and 28 km might be reasonable for the background aerosol conditions. However, for the case with a dense aerosol cloud like after the Hunga eruption, the line-of-sight optical depth becomes incredibly high. This means that the measurements at lower tangent points are not sensitive to changes at those altitudes, and the signal rather comes from upper levels that lie closer to the instrument. Under those conditions, instead of focusing on improving retrievals in places where the measurements are the most sensitive (based on K), the algorithm is pushed to retrieve hard in places with no sensitivity.

We agree, the convergence criteria we use might be sub-optimal for some situations but they ensure a reasonable convergence speed and a satisfactory quality of the results for the majority of the runs. In our opinion the only drawback of using suboptimal convergence criteria is a high number of iterations. No overall quality drawback is expected from this as no displacement of the retrieval focus at each particular iteration occurs.

 I am not sure that the algorithm with 100 iterations can be used in the operational environment. Can you plot a histogram showing the number of iterations under background conditions and under perturbed conditions (like volcanic eruptions or wildfires)?

Our algorithm has been developed as a research exercise and certainly needs an optimization to be used for a near real time processing. This optimization is however relatively straight forward. For now, we can easily process the entire data set within a few weeks using high performance computing facilities. The requested histograms are now provided in the supplement.

 The measurement noise can be quite different between LP and SCIAMACHY. I would not extrapolate conclusions derived from the analysis of SCIAMACHY spectra to OMPS LP.

We agree with the reviewer's comment and added the following comment to the manuscript text: "Although, results from SCIAMACHY cannot be directly transferred to other instruments, a degradation of the measurement quality with an increasing tangent height is rather common for limb-scatter observations."

 What do the horizontal green lines at 9 km represent? Does it mean that positive differences for v1.09 switch to negative below 9 km? If I interpret the error bars correctly, the standard deviation for differences is larger than +/-100% at lower altitudes (depending on latitude zone). How meaningful are the comparisons with a standard deviation greater than 100%?

Yes, these lines mean a switch between positive and negative differences. The reason for this is that V1.0.9 retrieves only down to 12.5 km and switches to a priori below. From the mathematical inversion theory point of veiw, signals with amplitudes below the noise level can still be detected. However, the obtained results should be handled with care.

Page 12, line 275: Have you described the collocation criteria for comparisons with SAGE III?

Indeed, we forgot to describe collocation criteria for the comparison with SAGE III. They are discussed at the beginning of Sect. 6 of the revised manuscript.

Page 14, Figure 7: Are you calculating zonal means from all available measurements for each instrument independently? If so, then the difference in the temporal and spatial sampling (particularly with SAGE III) can produce biases that are not accounted for.

Yes, monthly zonal means are calculated from each instrument independently. We agree this can introduce biases. The usage of colocated measurements is, however, disadvantageous for a time series comparison as the sampling of the measurements is strongly reduced. As the end users are rather interested in monthly mean climatologies [1, 2], we think our comparison is appropriate. Our internal tests show that general conclusions remain the same when collocated measurements are used in the comparison.

Page 14, lines 294-295 and Figure 7: OMPS LP aerosol retrievals at 869 nm were converted to 750 nm to compare with OSIRIS. Since this study validates OMPS LP retrievals, it would be better to run comparisons at the "native" wavelength and rather convert OSIRIS extinction to 869 nm.

As OMPS-LP data described in this paper are used to create a climatological data record of stratospheric aerosol extinction coefficients at 750 nm [2], we prefer to keep the comparison at this wavelength.

Page 15, lines 330-333: Why do you use 2022 data when you stated above that OSIRIS data quality degraded in 2022? Or is it a typo?

This is indeed a typo. We changed the range to 2012 – 2021.

Page 16, line 335: Both OSIRIS and LP have relatively dense sampling, so I am not sure what you meant here. Can you please elaborate on this?

Both instruments cannot measure in the darkness which leads to gaps at high latitudes in winter. As a result of the difference in the times of the equator crossing for OSIRIS and OMPS-LP orbits, the gaps are larger for OSIRIS.

Page 16, line 336: I agree that the cloud correction is one of the many factors that contribute to reduced quality of aerosol data and larger differences in the troposphere, but I will not say that it is the only one.

We think there is a consensus in the limb-scatter community that clouds is a major issue for the aerosol data comparisons in the troposphere. Certainly it is not the only reason for differences but this reason is enough to spoil the comparison if clouds are treated in different ways.

*Multi-panel figures: Please add labels (a, b, c, etc.) on all figures that have multiple panels.*

The labels have been added and the manuscript text has been adjusted to reference the panels in accordance with their labels.

*Figure 10, legend: What do you mean by "Relative mean differences"? Do you calculate zonal means first and then calculate the difference between the two monthly zonal means? Then it should be "Relative differences." Otherwise, clarify that in the text.*

We agree, these are relative differences.

*Section 7: There have been many publications in the last two years that describe the transport of volcanic aerosol after the Hunga eruption, which are not acknowledged here. Is there any reason for that? How do the conclusions of this study agree with previously published results?*

We added a paragraph at the end of Sect. 7 comparing our results to those from other studies.

*Minor comments: Page 2, line 46: it's not clear from the context what "this range" refer to. It might be better to say "... to the aerosol at the normalization altitude".*

Corrected as suggested by the reviewer.

*Page 2, line 46: It doesn't sound right when you state that the knowledge is the major source of uncertainty. Perhaps, "the lack of knowledge" or "incomplete knowledge".*

Changed to "a lack of the knowledge"

*Page 8, lines 213: should "for example".*

"example" is meant here as an adjective which belongs to "measurements", the construction "an example measurement" is correct.

*Page 8, line 214: should be "with the tangent point ground coordinates"*

Changed

*Page 8, line 216: should be "every third AK"*

Changed

*Page 11, line 258: the word "tangent" is used twice.*

Corrected

**References**

[1] Kovilakam, M., Thomason, L. W., Ernest, N., Rieger, L., Bourassa, A., and Millán, L.: The Global Space-based Stratospheric370 Aerosol Climatology (version 2.0): 1979 –2018, Earth Syst. Sci. Data, 12, 2607–2634, https://doi.org/10.5194/essd-12-2607-2020, 2020.

[2] Sofieva, V. F., Rozanov, A., Szelag, M., Burrows, J. P., Retscher, C., Damadeo, R., Degenstein, D., Rieger, L. A., and Bourassa, A.: A Climate Data Record of Stratospheric Aerosols, Earth Syst. Sci. Data Discuss. [preprint], https://doi.org/10.5194/essd-2023-538, in review, 2024.

---

## Author Comment (AC2)

We thank the reviewer for carefully reading our manuscript and helpful comments. Below, the reviewer's comments are marked in blue and our answers to the comments are written in black.

The authors present a new retrieval scheme for aerosol extinction coefficient derived from profiles of limb-scatter radiance as applied specifically to OMPS-LP data. Advantages over their previous algorithm are shown in a convincing manner. While there are several differences between the two algorithm versions, the authors state the overruling factor is the normalization approach. The manuscript would benefit from a sensitivity analysis for typical error sources, even a limited one would be enlightening. Highlighting results from the Hunga eruption is very nice. I recommend publishing this article, after successfully addressing the comments below.

A thorough investigation of the major error sources typical for retrievals of the aerosol extinction coefficients from limb-scatter measurements was published in [1]. Most of the results are applicable to our algorithm as well. The most significant error source is the assumption about the aerosol particle size distribution. A detailed investigation of its influence is a subject of a follow-up paper.

Is there a missing affiliation in the list on the cover page, i.e. 3?

corrected

Lines 136-144: Need to include quantitative estimates of the retrieval errors.

Unfortunately, the is no reliable way to quantify potential errors related to the absolute calibration, which are systematic errors. Where systematic errors have been identified, they are corrected for during the Level 1 calibration procedure. As we explain in the text, up to the present there are no indications of unknown errors in the absolute calibration of OMPS-LP level 1 data. If present these would lead to bias and propagate into the level 2 data product.

Section 4: Should make it clear that there are only a finite number of wavelengths to use since OMPS-LP does not download the full spectrum. That would be why you are not suggest using slightly different wavelengths to reduce interference from atmospheric absorption/emission.

We added the following sentence to the manuscript text in Sect. 4: "It might be advantageous to slightly shift the central point towards the shorter wavelengths. However, due to a sparse spectral sampling of the OMPS-LP level 1 data, this is not possible without including the water vapor band on the short-wavelength side."

Line 200: Should be $O_2$-A instead of $O_2$-B and wavelength is 688nm (Newnham & Ballard, 10.1029/98JD02799)

Indeed, we mixed up the A and B bands of $O_2$. This mistake is corrected in the revised manuscript. The wavelength of the $O_2$-B band is corrected to 688 nm.

**Line 204: $O_2$ has a band near 867 nm**

This band is not identified in the limb spectra shown in Fig. 1 and thus can be considered irrelevant.

**Figure 3: What solar irradiance spectra are used for these cases?**

The sun-normalized radiances are provided in OMPS-LP Level 1 data. These radiances are obtained using the solar irradiance spectra measured by the OMPS-LP instrument.

**Figure 3: What do the aerosol profiles look like with the new algorithm for altitudes above 35km? This should help the OMPS team know how well the stray light correction scheme performs.**

We added a figure illustrating the results for altitudes above 35 km to the supplement.

**Line 244: The normalization range does seem to be too low in altitude. If the solar normalized radiances up to 50 km are good enough to estimate surface albedo, then the normalization range for the V1.0.9 retrieval should be raised, maybe above 45 km.**

We would like to point out that in V2.1 all tangent heights are used to retrieve the effective surface albedo rather than one particular. From the results of this study, we cannot say if single tangent heights above 50 km are good enough to estimate the surface albedo. When optimizing the parameters for V1.0.9 we considered an option to raise the reference tangent height and found it sub-optimal with the setup of V1.0.9. There is certainly a potential to develop another retrieval which uses a higher reference tangent height but this is outside the scope of the current study.

**Line 253: What are the reflectance values for the various cases?**

We provided the obtained values in the supplement.

**Line 254: "scaling of the a priori at the ...reference tangent height" This appears to behave differently than trace gas retrievals from UV/VIS backscattered sunlight that use a normalizing spectra obtained from backscattered radiance spectra over a reference sector, typically a region with low trace gas amounts. The trace gas amounts from the reference sector are subtracted from the total trace gas.**

Yes, it behaves differently as the reference sector method works on the Level 2 data while here the subtraction of radiances takes place.

**Line 332: Is the same mean added to all three datasets to make the left panel of Fig. 8?**

Its own mean is added to each dataset. This is clarified in the text of the revised manuscript.

**References**

[1] Rieger, L. A., Malinina, E. P., Rozanov, A. V., Burrows, J. P., Bourassa, A. E., and Degenstein, D. A.: A study of the approaches used to retrieve aerosol extinction, as applied to limb observations made by OSIRIS and SCIA-MACHY, Atmos. Meas. Tech., 11, 3433-3445, https://doi.org/10.5194/amt-11-3433-2018, 2018.

---

## Author Comment (AC3)

We thank the editor for carefully reading our manuscript and helpful comments. Below, the reviewer's comments are marked in blue and our answers to the comments are written in black.

The stratospheric aerosol layer plays a critical role in the Earth's climate system through its impact on radiation, chemistry and the hydrological cycle. Impacted by large volcanic eruptions, its composition and loading can also reflect the influences of sulfur precursor emissions (SO2, OCS, DMS), extreme wildfires through Pyro-convection and the Asian Summer Monsoon transport pathways. The stratospheric aerosol layer has been studied since more than 4 decades through satellite-based solar occultation techniques, ground-based lidar and balloon-borne observations. More recently, limb observations have shown its ability to study stratospheric aerosol despite some limitations on calibration procedures, resolving complex radiation influence from scattering and absorbing and underlying assumptions on aerosol size distribution. Rozanov et al. (2024) utilizes the Ozone Mapping Profiler Suite- Limb Scatter instrument to study stratospheric aerosol extinction quasi-globally since 2012. Improvements of the retrieval algorithms are discussed in this paper and the results are compared with SAGE III/ISS and OSIRIS satellite observations. Overall, the strengths and limitations of the new algorithm are well exposed and convincing. This is a well-written, logically-structured and organized paper which merits to be published in AMT after some minor corrections can be applied and additional explanations could be provided.

1) L15P1: Solomon et al. 2011 do not report the presence of large amount of aerosols but rather an increase of stratospheric aerosols from moderate but frequent volcanic eruptions as reported by Vernier et al. (2011). I would recommend correcting this sentence.

The interpretation of the word "large" certainly depends of the reference. If the reference are post-Pinatubo conditions then the aerosol load considered by Solomon et al. 2011 is certainly not large. If one compares to background conditions around the year 2000, it is still large enough. To avoid any confusion here, we replaced "large" by "increased".

2) L25-26P2: Evan et al. 2023 report ozone loss soon after the HTHH eruption with limited explanations about the causes. Zhu et al. (2023) found that enhanced chlorine from marine sources was likely responsible of the ozone loss more than a week after the eruption rather than dynamical processes. The same study evokes a different ozone loss mechanism than traditional volcanic eruptions. I believe that some nuances could be made here.

We were not aware of Zhu et al. (2023) paper. Thank you for this hint. The information is added to the first paragraph of the introduction.

3) P2L29: This statement should be nuanced and is not fully correct. SAGE has provided quasi-global observations since 1979 but at rather low spatial sampling (30 profiles per day)

We think the statement "the availability of information .... is quite limited" is still correct, as SAGE sampling is not enough e.g. to create a latitude-longitude resolved climatology. Even considering limb-scatter and lidar instruments the information is still limited. We agree, however, that long-term measurements from SAGE instruments need to be acknowledged. To this end, we added the information suggested by the editor into the 6-th sentence of the paragraph, where we discuss occultation measurements.

4) P2L35: While describing SAGE data, some information regarding the fact that the spectra are self-calibrated through exo-atmospheric measurements might be of interest for the reader.

The information suggested by the editor is added to the text.

5) P3L62: I do not believe that this paragraph justifies well why CALIPSO is not used. As a matter of fact, I would recommend using the new stratospheric aerosol product level 3 developed recently (asdc ...). It could be used to understand the performance of OMPS algorithm when other datasets are not available (e.g. SAGE III/ISS in the polar winter regions Or near the tropopause where the variability of aerosol might be important and the influence of cirrus clouds in the tropics significant).

We think the objectives of the paper, which are the presentation of the retrieval and initial validation, are achieved using the two reference data sets (SAGE III and OSIRIS). Inclusion of CALIOP data would require a major rewriting of the paper, blow up its length and defocus the study. However, we agree with the editor that CALIOP data might be useful for upcoming studies focused on specific ranges of the atmosphere. We realized that the sentence about CALIOP data might be misunderstood as a total refusal to use these data in the comparisons. We re-formulated this sentence to highlight that we just prefer other data sources for this particular study to avoid potential ambiguity in the interpretation of the results and keep the door open for further comparisons.

In addition, I could not find how OMPS and other measurements were collocated with OSIRIS and SAGE III/ISS. Could you please clarify this

Indeed, we forgot to list the collocation criteria for the comparisons with SAGE III. They are now presented at the beginning of the Sect. 6. As stated in the third paragraph of Sect. 6, comparisons of the time series from OMPS-LP, OSIRIS and SAGE III are done using monthly zonal mean data (not collocated data).

6) P4L107: This is extremely difficult to make sense of this for non-specialist. I recommend to use some references but also to provide additional information by trying to avoid employing too many technical terms. Maybe a schematic describing the different steps of the algorithm could be useful here. Additional effort should be made here to further explain the different steps of the algorithm

The retrieval is based on the theory presented by [1], which indeed might be quite difficult to understand for a non-specialist. The work of [1] introduces some basis terms, which cannot be easily avoided. An attempt to avoid terms commonly used in the community is associated with a risk to make a description confusing even for readers who are familiar with the basics of the approach. A schematic diagram would not make much sense as the retrieval consist of one single step. All data are inverted at once without any intermediate steps. To clarify the issue we added an introductory paragraph at the beginning of Sect. 3 and some references.

**References**

[1] Rodgers, C. D.: Inverse methods for atmospheric sounding: Theory and practice, World Scientific, 2000.